# Global correlates of terrestrial and marine coverage by protected areas on islands

David Mouillot [1,2✉], Laure Velez[1], Eva Maire [1,3], Alizée Masson[1], Christina C. Hicks[3], James Moloney[4] & Marc Troussellier[1]

Many islands are biodiversity hotspots but also extinction epicenters. In addition to strong cultural connections to nature, islanders derive a significant part of their economy and broader wellbeing from this biodiversity. Islands are thus considered as the socio-ecosystems most vulnerable to species and habitat loss. Yet, the extent and key correlates of protected area coverage on islands is still unknown. Here we assess the relative influence of climate, geography, habitat diversity, culture, resource capacity, and human footprint on terrestrial and marine protected area coverage across 2323 inhabited islands globally. We show that, on average, 22% of terrestrial and 13% of marine island areas are under protection status, but that half of all islands have no protected areas. Climate, diversity of languages, human population density and development are strongly associated with differences observed in protected area coverage among islands. Our study suggests that economic development and population growth may critically limit the amount of protection on islands.

[1] Univ Montpellier, CNRS, Ifremer, IRD, MARBEC, Montpellier, France. [2] Australian Research Council Centre of Excellence for Coral Reef Studies, James Cook University, Townsville, QLD 4811, Australia. [3] Lancaster Environment Centre, Lancaster University, Lancaster, UK. [4] College of Science and Engineering, James Cook University, Townsville, QLD 4811, Australia. ✉email: david.mouillot@umontpellier.fr

Overexploitation of resources, climate change, and land-use intensification are among the most prevalent threats to biodiversity[1,2] with severe consequences for ecosystem functioning and human well-being[3,4]. Protected areas (PAs) are clearly defined geographical spaces, recognized, dedicated, and managed through legal or other effective means to achieve the long-term conservation of nature with associated ecosystem services and cultural values (IUCN 2008). PAs thus play a key role in achieving the Sustainable Development Goals (SDGs)[5]. Indeed, PAs have been shown to support species abundances[6,7], human well-being[8–10], adaptation to climate change[11], and efforts to alleviate poverty[12,13]. However, in many settings, access restriction imposed by PAs may negatively impact the livelihoods and food security of local users[14,15], may fail to adequately decrease human pressure within PA boundaries[16–18], or may displace, and in some cases increase, human pressure into other areas beyond PA boundaries inducing a so-called "leakage" of environmental degradation[19,20]. The balance and distribution of fortune and misfortune (e.g., benefits vs. costs) among PA users may explain why the establishment of PAs is still lagging and poorly accepted in some places compared to others[21]. If international efforts and political commitments to scale up PA coverage are to be realized[5,22,23], they will need to be built on a better understanding of the social and environmental factors that promote or inhibit the creation of PAs. However, many factors are potentially at play and we lack a comprehensive understanding of how these factors determine whether, where, and the extent to which terrestrial and marine areas are currently protected or are likely to be in the near future.

On islands, perhaps more than anywhere in the world, natural and human systems share a history of strong interdependence whereby, with few exceptions, biodiversity has supported long-term human well-being[24]. In addition to strong cultural connections to nature[25], islanders derive a significant part of their economy and broader well-being from a wealth of natural resources with biodiversity-based tourism and fisheries accounting for more than half of the GDP in small island developing states[26]. However, food production is a critical issue on many islands where land area is limited and human populations are increasing, not only challenging their sustainable development but also accelerating the conversion of available natural land to crop production[27] with major consequences for ecosystems[28]. Human-mediated predator invasions have also contributed to many species extinctions on islands[29], particularly because when native biota have not co-evolved with predators, they have not developed escape behaviors and are vulnerable to predation[30]. Therefore, islands, accounting for up to two-thirds of recent species extinctions[31], can be considered the most vulnerable socio-ecosystems to biodiversity loss[32]. Yet, the extent to which insular terrestrial and marine areas are covered by PAs compared to their continental counterparts is virtually unknown, while the key correlates of insular protection efforts are yet to be revealed.

A number of international agreements and targets seek to safeguard ecosystems and their biodiversity. Notably, the Convention on Biological Diversity (CBD) legally commits governments to conserve biodiversity through a strategic action plan. The Aichi targets are a key element of the CBD, with Target 11 to protect at least 17% of terrestrial and 10% of marine areas by 2020. Yet, despite accelerating conservation efforts in the past decade and the establishment of large-scale PAs[21,33], these 2020 Aichi targets are unlikely to be achieved in most countries and will likely be renegotiated[34,35]. Moreover, these country-scale targets are widely viewed as insufficient or inadequate to halt the ongoing loss of species[36,37]. Indeed, the rate of wilderness loss is nearly double the rate of increase in protection of terrestrial areas[38] and more than half of the oceanic areas are now exploited[39]. A new call to protect at least 30% of the Earth's surface by 2030 has been proposed by a broad coalition of environmental organizations as a milestone toward the more ambitious target of protecting half of the planet by 2050[23,40]. Although the legal designation of a PA does not guarantee effective species and habitat protection[16,17,41,42], these targets and measures of achievements will likely remain and be complemented by evaluations of the broader PA benefits[34,35].

To study global PA coverage on islands, we first identify all inhabited islands, located at least 10 km from the closest continent, with a minimum surface area of 10 km² and a maximum of 2,166,000 km² corresponding to Greenland, that is, the largest island on Earth (see "Methods"). Next, for each island, we quantify the amount of terrestrial and marine area (continental shelf in territorial waters within 12 nautical miles) currently considered within a PA. Then, we estimate: (1) the global extent and heterogeneity of terrestrial and marine PA coverage on islands; (2) the proportion of islands that currently meet the Aichi Target 11 of the CBD, to protect 17% of terrestrial and 10% of marine areas by 2020, and (3) the proportion of islands that currently meet the 2030 target of protecting 30% of terrestrial and marine areas. Finally, to understand how social and environmental factors are associated with PA coverage on islands, we model (i) the probability of terrestrial and marine PA presence on each island, and (ii) the likelihood that each island already meets the new global target of 30% coverage on both terrestrial and marine areas. Above 30% coverage, we consider that modeling differences among islands to be of less importance. Our main goal is not to focus on high values of protection coverage (30–100%), but to investigate the correlates of achievement of different thresholds corresponding to policy targets. We show that, on average, islands are more covered by PAs than the global coverage but with a high heterogeneity. This heterogeneity of protection coverage among islands is well explained by a small set of social and environmental factors. We suggest that low economic development and high population growth may limit the amount of protection on islands.

## Results

**Global protection coverage on islands**. We identified 2503 inhabited islands globally with a mean surface area of 3224 km² (SD = 48,653 km²). Information on key social and environmental factors were missing for a number of high latitude islands (e.g., arctic ocean), we therefore retained 2323 islands in our analyses; Greenland was excluded with this criterion (see "Methods"). These 2323 islands belong to 101 different countries with a mean of 23 islands per country (SD = 60 islands).

We estimated the average PA coverage on an island to be 22% for terrestrial (SD = 37%) and 13% for marine (SD = 27%) areas. These values exceed both Aichi Target 11 (17% terrestrial and 10% marine areas protected by 2020) and are greater than current global averages (15% terrestrial and 7% marine areas are protected). However, variability across islands is high; the global distribution of PA coverage is U-shaped with roughly half of islands having either no terrestrial or no marine PAs (Fig. 1a). Specifically, 57% of islands (n = 1361) have no terrestrial PAs and 42% (n = 1007) have no marine PAs, while 14% (n = 335) and 5% (n = 110) of islands have over 90% of their terrestrial or marine areas covered by PAs, respectively (Fig. 1a). Twenty-nine percent (n = 664) of islands currently meet the 2020 Aichi Target 11 for terrestrial (>17%) coverage and 24% (n = 560) for marine (>10%), while only 15% (n = 368) of islands meet both (Fig. 1a). Twenty-five percent (n = 574) of islands currently meet the 2030 target of 30% of terrestrial areas under protection and 16% (n = 372) of islands for marine areas, whereas 11% (n = 249) of

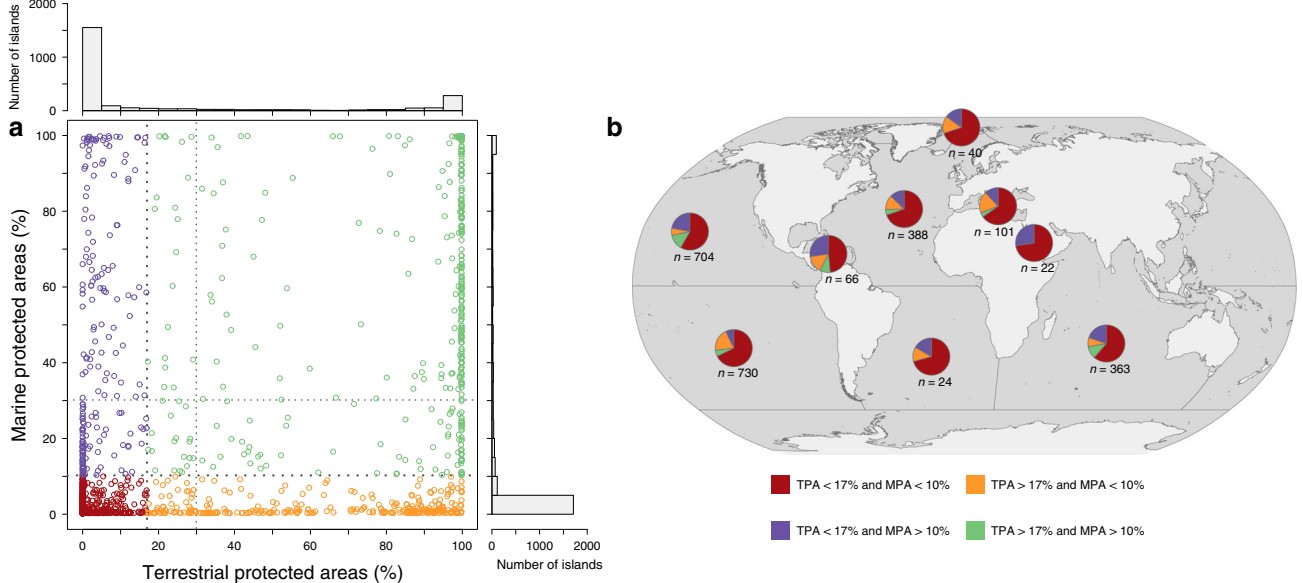

**Fig. 1 Spatial coverage of terrestrial and marine protected areas on the 2323 inhabited islands across oceans.** Global relationship between terrestrial and marine protected area coverage on islands. **a** The islands that meet 2020 conservation targets (>17% on land and >10% on sea; Aichi targets) are colored in green; islands meeting only the target on land in orange; islands meeting only the target on sea in purple; and islands meeting none in red. The 2030 conservation target (30% coverage) is also indicated by dashed lines. Spatial coverage by protected areas is significantly correlated between terrestrial and marine areas (Pearson's coefficient of correlation $r = 0.5$, two-sided $p < 0.001$). Proportions of islands meeting both one or none 2020 conservation target in each ocean **b**, and colors represent the same categories as in **a**.

islands meet this target for both terrestrial and marine areas (Fig. 1a).

Overall, the relationship between the coverage of terrestrial and marine PAs on islands is relatively weak ($r = 0.5$), but significant ($p < 0.001$) owing to the high number of islands having little or no PA coverage in both terrestrial and marine areas (Fig. 1a).

The spatial distribution of PA coverage is highly heterogeneous among oceans (Fig. 1b) with the Caribbean, North Pacific, and Indian Oceans showing the highest proportion (>10%) of islands meeting both 2020 conservation targets, while the North and South Atlantic, the Arctic, the Red Sea, and the Mediterranean Sea have the highest proportion (>70%) of islands meeting neither 2020 conservation target (Fig. 1b). Across terrestrial areas, the Mediterranean is the region with the highest proportion (>30%) of islands achieving the 2020 conservation target (>17% coverage), whereas across marine areas, the Caribbean Sea, the North Pacific, and Indian Ocean are the regions with the highest proportion (>30%) of islands achieving the 2020 conservation target (>10% coverage).

**Modeling the presence of any PAs.** We used binomial generalized linear models (GLMs) to explore how 16 social and environmental factors are associated with the presence of any PAs in terrestrial and marine areas for each island (binary response variable 0/1), independently of coverage. These factors refer to climate conditions, aspects of island geography, habitat diversity, island culture, resource capacity, and human footprint (Table 1) (see "Methods" for more details). Maximum and minimum temperature and precipitation reflect island climates; distance from the mainland, average elevation, and area covered by freshwater capture aspects of island geography; total surface area, maximum altitude, and fractal dimension provide an indicator of habitat diversity; number of languages and whether institutional languages are indigenous, non-indigenous, or mixed capture elements of island culture; human development index (HDI) and sovereignty provide indicators of resource capacity; and population density and cropland area capture human footprint.

Rationale, hypotheses, and expectations for each factor are detailed in Table 1.

The presence of any PAs on islands is well explained by ten retained factors as indicated by a high area under the curve (AUC) value for terrestrial (0.79) and marine (0.81) models (see Table 2 for model details). HDI has by far the strongest effect ($R_E^2 = 32\%$) for both terrestrial and marine areas (Fig. 2a, b). By contrast, human footprint (population density and cropland area cumulated) has the lowest impact on both terrestrial ($R_E^2 = 2.5\%$) and marine ($R_E^2 = 2.3\%$) PA presence on a given island.

For both terrestrial and marine areas, the probability of having any PA coverage has a quadratic (hump-shape) relationship with minimum annual temperature (the peak is ~0–5 °C) and maximum annual precipitation (the peak is ~700 mm), so being the highest for tropical and sub-tropical areas (Fig. 3a–d). The majority of islands without any PAs are located either in cold-climate regions (i.e., minimum annual temperature <−20 °C) or in equatorial areas (i.e., minimum annual temperature >15 °C and maximum annual precipitation >1000 mm). The probability of having some PA coverage increases with island surface area (Supplementary Fig. 1e, f) and decreases with distance from the mainland for both terrestrial and marine areas (Fig. 3e, f). The fractal dimension of the island positively influences the probability of having some PA coverage in both terrestrial and marine areas with more tortuous islands more likely to have some protection (Fig. 3g, h). The total number of languages spoken in the country and HDI are strongly, monotonically, and positively related to the probability of having some PA coverage for both terrestrial and marine areas; islands with a lower national HDI or fewer national languages were more likely to have no PAs than islands with a higher HDI or more languages (Fig. 3i, j, m, n). Finally, the nature of institutional languages (indigenous vs. non-indigenous) also has a link with the presence of PAs. Islands where all institutional languages are indigenous are more likely to have marine PAs than islands where institutional languages are mixed or all are non-indigenous (Fig. 3l). For terrestrial areas, the opposite pattern appears; islands where all institutional languages

**Table 1 Main rationale and hypotheses explaining the expected relationships between the 16 factors used in our study and the presence or the 30% coverage of protected areas on islands for both terrestrial and marine areas.**

| Factor (abbreviation) | Rationale and hypotheses | Expected relationship with protection coverage |
|---|---|---|
| Maximum temperature (Max temp) | The latitudinal biodiversity gradient is related to air and seawater temperature with a greater diversity of species found in the tropics[47,48,60]. Islands with high temperature are likely to host more species and thus expected to have greater protection | Positive |
| Minimum temperature (Min temp) | Many species that originated from the tropics cannot cope with freezing conditions given climate niche conservatism[82]. So, islands with low minimum temperatures are likely to host fewer species and thus set lower levels of protection | Positive |
| Maximum precipitation (Max prec) | Precipitation is essential for forest and wetland habitats where most of terrestrial biodiversity is found[83,84]. Islands with more precipitation are likely to host more species and thus expected to have greater protection | Positive |
| Minimum precipitation (Min prec) | Drought severely limits life on terrestrial ecosystems[85,86]. Islands with dry conditions are likely to host fewer species and thus expected to have lower levels of protection | Positive |
| Freshwater area | Freshwater is necessary for most terrestrial species and is critical for agriculture and human livelihoods[87]. Islands with large areas covered by freshwater are likely to have a greater diversity of species and thus expected to have more protection. However, such islands are also likely to have more productive land available for agriculture creating competition for space and thus the expectation of lower levels of protection | Positive or negative |
| Surface area (Surf area) | Large islands host more varied habitats, and consequently more species[86,88], but also have more space for human activities (cities, agriculture, fisheries) and thus less competition for space. Large islands are expected to have more protection and to achieve minimum protection coverage more easily. However, small islands are expected to be more likely to reach ambitious conservation targets because their areal coverage requirements are lower | Positive for the least demanding conservation target (presence of protection) but negative for the most demanding (>30% coverage) |
| Distance to the mainland (Dist to mainland) | According to the theory of biogeography[89], distance to the mainland is the main driver of species richness on islands, so isolated islands may host less species and thus receive lower levels of protection. Besides, islands located far from the mainland have greater resource management requirements (e.g., cost of establishing and running protected areas). These islands are thus expected to have less protection[90]. An "Islandness" feeling is likely to increase with distance from the continent so isolated islands may reject restrictions more strongly than connected islands and such conflicts may decrease protection area acceptance and protection coverage[64]. Yet, we cannot preclude some exceptions if biodiversity is the main richness of small and isolated islands and a critical source of economic incomes (tourism) that need protection | Negative |
| Maximum altitude (Max alt) | Altitude can create a wide variety of habitats and environmental niches but also isolated populations[91]. Mountainous islands are likely to host more species, particularly endemics, and thus expected to receive more protection | Positive |
| Mean altitude (Mean alt) | A high mean altitude can be detrimental to agriculture and urbanization (e.g., colder conditions)[92]. High altitude islands are likely to host fewer people and activities reducing competition for space and are thus expected to have more protection[51] | Positive |
| Fractal dimension (Fractal dim) | A tortuous coastal line may provide more terrestrial and marine habitats and are thus likely to host more species[93]. Furthermore, tortuous islands have less favorable conditions to host large cities and agriculture, reducing competition for space, and are thus expected to have more protection | Positive |
| Population density (Pop density) | High human population densities are associated with overexploitation of natural resources[7] or more degraded habitats[94], and create competition for space[51]. Densely populated islands are thus expected to have lower levels of protection | Negative |
| Cropland area | | Negative |

**Table 1 (continued)**

| Factor (abbreviation) | Rationale and hypotheses | Expected relationship with protection coverage |
|---|---|---|
| | Under demographic pressure, many islands have undergone a transition from food surpluses to deficits as land was converted for cash crops by a land-owning elite[27]. Moreover, cropland area is in direct competition with other uses, such as protected areas, for space[72]. Islands with extensive cropland area are expected to have lower levels of protection | |
| Number of languages (No. of languages) | The number of languages spoken in a country or island is associated with the diversity of landscapes and cultures. The number of languages is also positively associated with species diversity[55,56]. Islands with a greater number of languages are expected to host more species and are thus expected to have more protection | Positive |
| Sovereignty | Islands that are territories, and not states, can have access to additional resources from the sovereign country (e.g., human and economic capital). Independent islands are therefore expected to have lower resource capacity and lower levels of protection | Negative |
| Nature of institutional languages (Inst languages) | Social capital, which tends to be greater where languages are common, can influence a countries institutional performance[62]. The nature of institutional languages (i.e., indigenous, non-indigenous, or both) reflects the historical colonization and its footprint. Social capital is therefore likely to be lower in islands with a mix of institutional languages (i.e., both indigenous and non-indigenous). Islands with a mix of institutional languages are expected to have lower capacity to reach shared agreement on management, and are expected to have less protection | Negative for a mix between indigenous and non-indigenous languages |
| Human development index (HDI) | Countries with a higher human development index (measure of health, education, and economy) are likely to have a greater capacity to manage their environment[43,60], and consequently these islands are expected to have more protection | Positive |

Only the four last factors are estimated at the country scale, the others are assessed at the island scale.

are non-indigenous are more likely to have terrestrial PAs (Fig. 3k). For both terrestrial and marine areas, islands where institutional languages are both indigenous and non-indigenous, so where a local and a colonial influence co-exist, are the least likely to have PAs. Partial regression plots of other factors are shown in Supplementary Fig. 1. We also conducted a sensitivity analysis repeating the analysis with "country" as a random effect to allow different intercept values for different countries (see "Methods"). We obtained very consistent results (Supplementary Table 1 and Supplementary Fig. 2), suggesting that, within each country, islands show differences in PA presence that are related to the same factors, although country-scale factors have lower AIC weights.

**Modeling the achievement of 30% protection coverage.** We again used binomial GLMs to explore how the same set of 16 social and environmental factors (Table 1) are associated with the current achievement of the 2030 conservation target for each of the 2323 islands (binary response variable 0/1 so success for reaching >30% protection coverage). For terrestrial PA coverage, the most parsimonious model retained 14 factors and performed well (AUC value = 0.77) (Table 2). Population density and HDI have the strongest effect with $R_E^2 = 17\%$ (Fig. 2c). When pooled together, human-related factors have a greater effect than environmental-related factors (60% against 40%), each of the latter accounting for less than $R_E^2 = 10\%$. For marine PA coverage, ten factors were retained in the best model, which also performed well (AUC value = 0.79) (Table 2). The most dominant factors associated with the achievement of 30% PA coverage are: cultural, the number of languages accounts for $R_E^2 = 22\%$; climatic, maximum temperature accounts for $R_E^2 = 19\%$, and

resource capacity, HDI accounts for $R_E^2 = 16\%$ (Fig. 2d). In contrast, factors related to geography or habitat diversity have the lowest dominance in this model with $R_E^2 < 5\%$.

We used partial regression plots to highlight the effects on achieving targets of the main social and environmental factors while controlling for the others (Fig. 4). On terrestrial areas, the probability of achieving >30% PA coverage increases with mean altitude, but decreases with cropland area (Fig. 4a, e), whereas on marine areas this probability strongly increases with fractal dimension or tortuosity (Fig. 4d). For both terrestrial and marine areas, population density has a negative influence on the 30% PA target achievement (Fig. 4g, h), whereas the number of languages has a positive effect (Fig. 4i, j). Islands with both indigenous and non-indigenous institutional languages are the least likely to have achieved 30% of PA coverage for both terrestrial and marine areas, while islands with only non-indigenous institutional languages are most likely to have achieved the 30% PA target (Fig. 4k, l). Finally, HDI is positively associated with PA coverage for both terrestrial and marine areas; islands with a higher HDI are more likely to meet the 30% PA target (Fig. 4m, n). However, this positive relation with HDI is saturating for terrestrial areas after 0.7 (only 20% of islands), while islands with the highest levels of HDI have the highest probability of meeting the 30% PA target for marine areas (up to 40% of islands). Partial regression plots of other factors are shown in Supplementary Fig. 3. We also conducted a sensitivity analysis repeating the analysis with "country" as a random effect to allow different intercept values for different countries (see "Methods"). We obtained very consistent results (Supplementary Table 1 and Supplementary Fig. 4), suggesting that, within each country, islands show differences in the achievement of 30% PA coverage that are

**Table 2 Results from binomial generalized linear models predicting the presence of protected areas and the achievement of the 2030 targets (30% coverage on both terrestrial and marine areas) for the 2323 islands globally as a function of 16 factors (see Table 1 and "Methods").**

| | | Presence of protection | | | | 30% of protection coverage | | | |
| --- | --- | --- | --- | --- | --- | --- | --- | --- | --- |
| | | Terrestrial protected areas | | Marine protected areas | | Terrestrial protected areas | | Marine protected areas | |
| | d.f. | AIC weight | F value | AIC weight | F value | AIC weight | F value | AIC weight | F value |
| Maximum temperature | 2 | 1 | 11.8*** | 0.30 | 10.8*** | 1 | 15.7*** | 1 | 39.8*** |
| Minimum temperature | 2 | 1 | 11.7*** | 0.99 | 29.8*** | 1 | 13.1*** | 1 | 11.5*** |
| Maximum precipitation | 2 | 1 | 9.7*** | 1 | 10.7*** | 0.98 | 5.8** | | 8.5*** |
| Minimum precipitation | 2 | 0.42 | | 0.99 | 19.6*** | 0.75 | 3.0* | 0.94 | 4.8* |
| Surface area | 2 | 1 | 13.3*** | 1 | 18.4*** | 0.45 | | 0.74 | 3.7* |
| Distance to the mainland | 2 | 0.96 | 5.2* | 0.40 | | 0.89 | 4.1* | 0.21 | |
| Maximum altitude | 2 | 0.61 | 0.6NS | 0.48 | | 0.96 | 4.0* | 0.20 | |
| Mean altitude | 2 | 0.45 | | | | 0.82 | 2.6* | 0.15 | |
| Fractal dimension | 2 | 1 | 10.7*** | 1 | 16.4*** | 0.42 | 2.1NS | 0.99 | 8.0*** |
| Cropland area | 2 | 0.49 | | 0.26 | | 0.63 | 1.3NS | 0.14 | |
| Freshwater area | 2 | 0.26 | 1.0 | 0.48 | | 0.54 | | 0.38 | |
| Population density | 2 | 0.20 | | 0.30 | | 1 | 22.7*** | 1 | 22.8*** |
| Number of languages | 2 | 1 | 15.0*** | 1 | 27.2*** | 1 | 15.5*** | 1 | 54.0*** |
| Institutional languages | 2 | 1 | 8.3*** | 1 | 14.1*** | 1 | 10.5*** | 1 | 9.7*** |
| Sovereignty | 2 | 0.38 | | 0.92 | 6.9* | 0.57 | 2.1NS | 0.29 | |
| Human development index | 2 | 1 | 99.5*** | 1 | 113.8*** | 1 | 33.6*** | 1 | 35.0*** |

d.f. is the degree of freedom for each factor, AIC weight represents the importance of each factor in the best models and the F value its influence on the predicted variable (NS, not significant, *$p < 0.05$, **$p < 0.01$, ***$p < 0.001$). Only factors retained in the most parsimonious models (according to a backward selection procedure based on AIC) are statistically tested.

related to the same factors, although country-scale factors have lower AIC weights.

## Discussion

We provide a global assessment of PA coverage on islands, utilizing an exhaustive database gathering information on 2323 islands. The average proportion of an island's area within a PA exceeds the 2020 country-based Aichi 11 conservation targets and is greater than the proportion of continental areas covered (22% compared to 15% for terrestrial areas and 13% compared to 7% for marine areas). However, there is considerable variability in PA coverage among islands globally. Nearly half the islands (>1000) have no terrestrial or marine PAs, whereas 335 islands have >90% of their terrestrial areas and 110 islands have >90% of their marine areas covered by PAs. We did not model this U-shape distribution to avoid focusing on extreme high values, but on certain thresholds that correspond to policy targets. Yet, the modeling of PA coverage on islands, as a continuous quantitative variable, using, for instance, zero-one inflated beta regressions with Bayesian inferences, may unravel another set of main factors and other relationships driven by islands with very high or even 100% coverage. These fully protected islands, often very small, were given low weight in our analyses. Many more small and uninhabited islands, which were not considered in our study focusing only on inhabited islands with a minimum size of 10 km², are certainly fully covered by marine and terrestrial PAs. Investigating the full range of island size would require another, even more ambitious, analysis with a very high spatial resolution for most factors (<100 m).

The global heterogeneity in PA coverage among islands is not surprising given previous results showing highly variable PA coverage among countries[21,43,44] and a disproportional effort, particularly boosted by the recent creation of very large PAs[21,33], in territories with few human uses[45,46]. Beyond these previous studies, we show that the variation in the presence and spatial extent of PA coverage can be explained by a limited set of social and environmental factors, among which only five are consistently retained in the most parsimonious models (Table 2): minimum annual temperature, maximum annual precipitation, the number of national languages, the nature of institutional languages, and HDI.

The close link between climatic variables and PAs can be a consequence of the latitudinal biodiversity gradient from polar to tropical ecosystems[47,48]. Many tropical islands are recognized as biodiversity hotspots given their level of endemism (e.g., Madagascar)[49,50], and have been an early focus of conservation efforts for a long time[51]. Thus, even if climate change will likely modify climatic conditions on most islands, there is no expectation these factors will promote the extension of insular PAs at least in a short or middle time period. Conversely, other biodiversity aspects like functional or evolutionary rarity[52,53] may be considered in the implementation of new PAs on islands.

The negative relationship between population density and protection effort is related to previous findings where human pressure has been an indicator of the threat to biodiversity[54]. This relationship is not surprising since the majority of land close to human settlements is dedicated to urban development or agriculture. For inhabited islands, and even more so for the smallest ones, this is an immense challenge to manage both nature conservation and human development, which at best allows only small PAs to be established. This may be one explanation for the difference in the importance of this factor between the models; human density is only a limiting factor to reach the 30% protection coverage target, but not for the presence of PAs. Likewise, the extent of cropland area only plays a negative role in the achievement of 30% PA coverage in terrestrial areas.

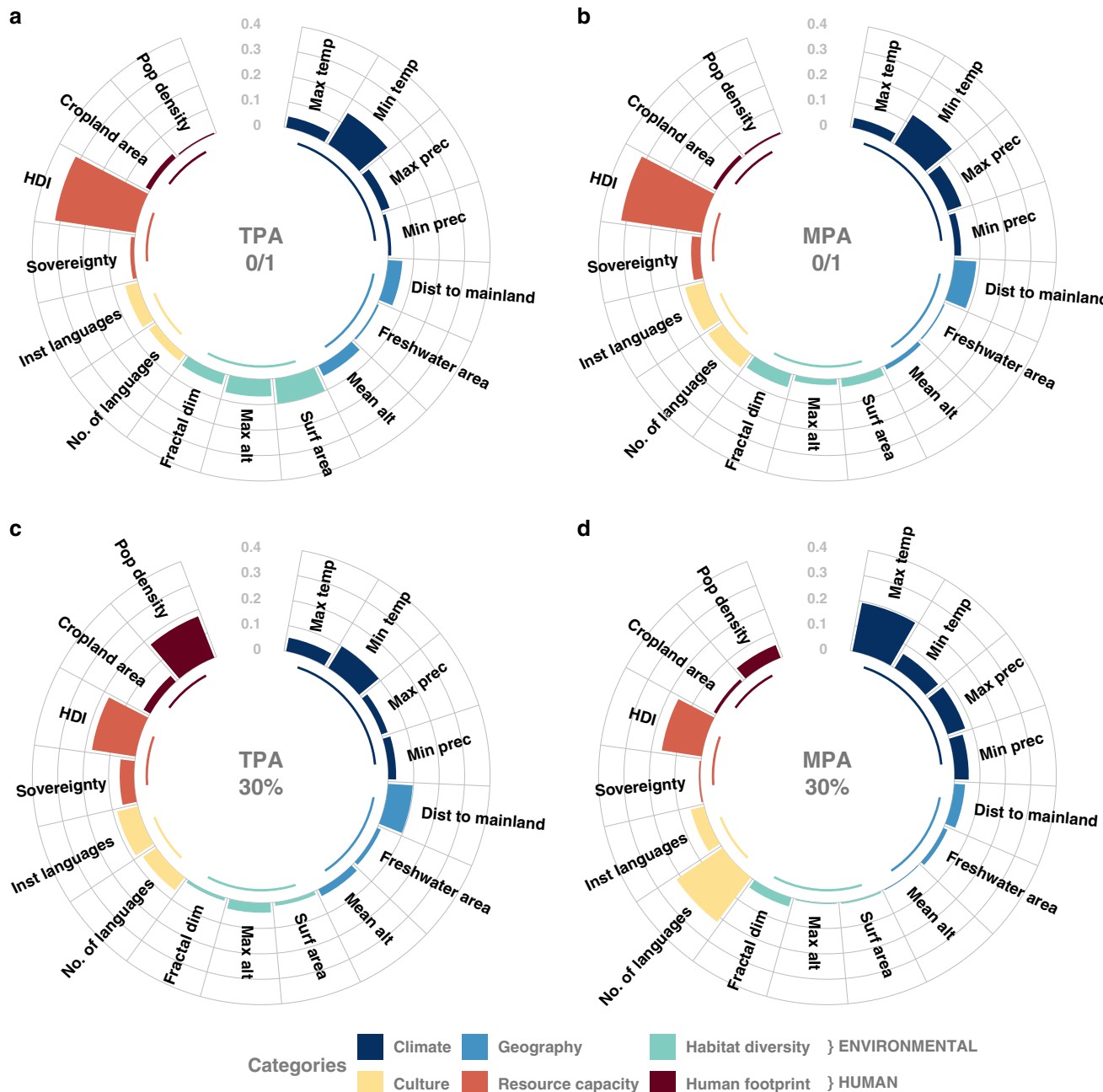

**Fig. 2 Relative dominance of environmental and human factors.** Circular plots showing the relative dominance of each factor in explaining the presence of protected areas on terrestrial (**a**) or marine (**b**) areas and the achievement of 30% coverage by protected areas on terrestrial (**c**) or marine (**d**) areas. Dominance values are based on the Estrella's $R_E^2$ index and sum to 1 for each model with a different color for each factor category. See Table 1 for factor abbreviations.

Areas where language diversity is greater tend to also possess greater ecological diversity[55,56] likely because of common environmental drivers rather than a causal link[57]. These multifaceted diversity hotspots are thus more likely to have been a focus of recent protection efforts, despite reported spatial mismatches[58,59]. However, these hotspots are currently experiencing the greatest rates of both species and cultural loss and should thus feature as priority areas for conservation[60,61]. Where both indigenous and non-indigenous institutional languages co-exist, barriers to communication may exist impacting a country's institutional performance[62]. Clear lines of communication, decision-making processes, and trust in public institutions are important for reaching an agreement in decisions over the management of

common resources. However, cultural diversity also generates different learning heuristics and perspectives, which may increase the ability of social groups to innovate and adapt to social change[63]. In areas where a mix of institutional languages exists, efforts to build confidence in public institutions and to broker communication and democratic decision making are likely to have a positive effect on PA coverage.

Island maximum altitude and fractal dimension have consistent positive relationships with PA coverage, suggesting that geography is a major constraint on conservation efforts with islands less attractive for land use by humans (e.g., agriculture) receiving more protection. Our results suggest that we are currently favoring the ease of PA establishment over the need for

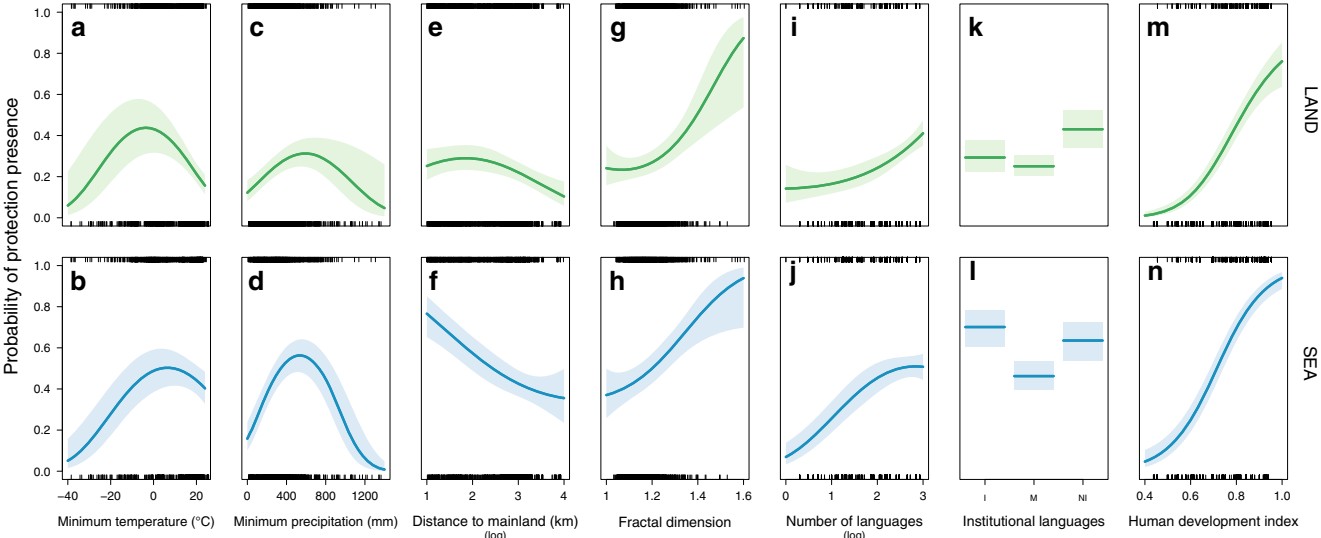

**Fig. 3 Relationships between the presence of protected areas on islands and the dominant factors.** Partial regression plots showing the influence of the seven main factors on the residual probability of having any coverage by protected areas (binary response presence/absence) on terrestrial (first raw, green curves) and marine (second raw, blue curves) areas of the 2323 islands globally, conditioned on the median value of all other retained factors. Institutional languages: I, indigenous only; NI, non-indigenous only or M, mix. The colored shaded areas are the 95% confidence intervals of the relationships.

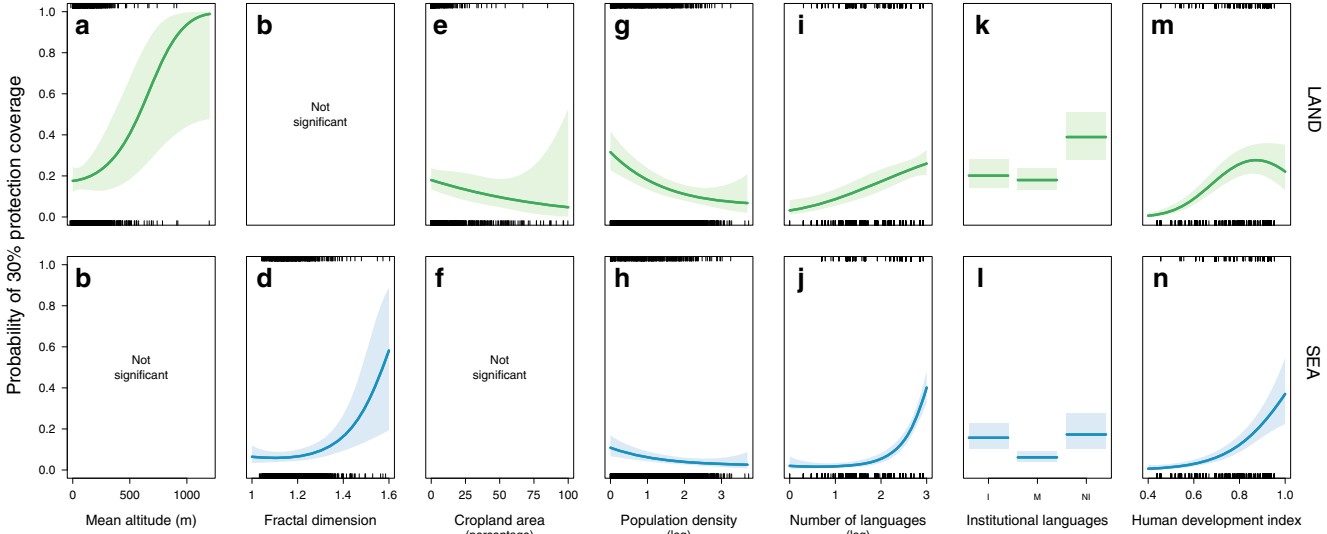

**Fig. 4 Relationships between the achievement of 30% coverage by protected areas on islands and the dominant factors.** Partial regression plots showing the influence of the seven main factors on the probability of reaching 30% protection coverage (binary response success/failure) on terrestrial (first raw, green curves) and marine (second raw, blue curves) areas of the 2323 islands globally, conditioned on the median value of all other retained factors. Institutional languages: I, indigenous only; NI, non-indigenous only or M, mix. The colored shaded areas are the 95% confidence intervals of the relationships.

protection in human-dominated landscapes[46]. The negative relationship between distance to the nearest mainland and the presence of PAs could be explained by the islandness feeling: when an island is located far away from a mainland, there are more difficulties to accept rules that restrict uses[64]. Isolated islands may also offer fewer alternatives to in situ natural resource exploitation to feed people and provide livelihoods, creating conflicts with protection efforts[65]. Another hypothesis may be that islands located far from the mainland do not have enough economic resources to create and manage PAs. For instance, the total cost of marine PA establishment ranges between $41 and $3,192,450 per km$^2$ [66], while the range of annual cost to run a terrestrial PA is estimated to between $0.1 and $1,000,000 km$^{-2}$ year$^{-1}$ [67]. This is supported by HDI being the predominant socioeconomic factor explaining the establishment of PAs[43]. Yet, remote islands of sovereign states may certainly have lower HDI than reported at the country scale, but this information was not available at the island scale. As an alternative, the decrease of HDI with island remoteness was certainly reflected by the distance to the nearest mainland in our models so accounted for in our partial plots of individual effects conditioned by the others (Figs. 3 and 4).

Recent studies show that PAs which provide economic benefits and support the well-being of local communities are more likely to promote efficient biodiversity protection[68]. Indeed, PAs on islands are likely to be inefficient if they are not inclusive, sufficiently sensitive to locality[26,64], or depend on wider law enforcement and limiting corruption[41]. PAs and conservation

measures or policy, often conceived in developed nations, may not be the right fit given a strong history of local management and tenure in developing island countries[14]. On these islands, there is a high risk of getting little return on investment in development to create terrestrial PAs. Consequently, efforts are galvanizing in support of other effective area-based conservation measures (OECMs), which may fare better in the near future, particularly in tropical and sub-tropical islands where they are likely to generate less conflict. For instance, co- and well-managed marine PAs have the potential to increase catches of commercial species in surrounding fishing grounds[69] but also support human well-being[9]. Sustainable fisheries can thus become an alternative to agriculture under climate change and land desertification. Unfortunately, spatial-based community management, tenured areas or OECMs are not currently represented in the World Database on Protected Area (WDPA) database, although some islands may prefer to adopt community-based management or multiple SDGs rather than pursuing western PA targets[5]. The factors driving OECM establishment or coverage are still unknown and may differ from those highlighted in this study.

Islands, if sufficiently protected, may play a key role in both biodiversity conservation and the achievement of multiple SDGs. Beyond extending the global coverage by PAs, the goal is to prioritize the right parts of Earth, that is, those concentrating small-ranged species[40]. Islands are at the forefront of these priority areas given their disproportional level of endemic and threatened species[31,70]. Islands may act as stepping stones to ensure species connectivity across large networks of marine PAs, which is critical under increasing fishing pressure and climate change to help species colonize favorable habitats and track their thermal preference[11]. However, PAs may fall short of halting biodiversity loss, even if their coverage increases, since perturbations can remain high inside their boundaries and even amplify outside their boundaries by leakage[16–19]. Moreover, protecting half the planet, or even 30%, may appear unrealistic because of the impact to food provision for one billion people[71,72]. PAs should be thus complemented by broader evaluations of their benefits, including species-targeted conservation actions and habitat restoration[34,73]. Yet, the future of economic development and population growth may critically determine the amount and efficiency of protection on islands so policies can make a decisive difference on the way to achieve the SDGs by promoting both nature conservation and human well-being.

## Methods

**Island selection**. To study global conservation efforts on islands, we based our selection on two criteria. First, suitable islands were identified as being at least 10 km² in area, due to the spatial resolution of explanatory variables, and the likelihood that these islands are inhabited, up to 2.17 million km² corresponding to Greenland, which is considered to be the largest island in the world (Australia is three times larger, but sits on its own continental plate, so is considered to be a continent), but excluded in this study due to a lack of data at high latitude. Second, the island polygon had to be located at >10 km from the nearest continent (Asia, Africa, North America, South America, Antarctica, Europe, and Australia) to minimize the direct socioeconomic influence of the mainland (through bridge or other proximate links).

We extracted the islands layer from the Global Administrative Areas—GADM v.2.8 (Global Administrative Areas 2016) spatial dataset, which maps administrative areas at a range of scales from national to local. GADM not only maps administrative areas at a high spatial resolution, but also for each area it provided attributes, such as the name and variant names. To aggregate the polygons into landmasses (islands and continents), we first dissolved all internal polygons, converted this output to the Behrmann (world) projected coordinate system for quantitative analyses, and then calculated the areas of all landmasses (km²). All polygons >10 km² but excluding the continents (see above) were selected. From this output, all islands beyond 10 km from a continent were extracted using the "select by location" function in ArcGIS 10.3. Finally, we intersected all polygons with the Gridded Population of the World, Version 4 (GPWv4) for the year 2015 and we only considered inhabited islands, resulting in a vector polygon dataset of 2503 islands.

**Terrestrial and marine coverage by PAs**. To determine the percentage of protection for each individual island, we estimated the proportion of terrestrial and marine area, the latter defined as the part of the continental shelf in the territorial waters (12 nautical miles), which intersected with the WDPA on September 2017. Continental shelf has been extracted from the GEBCO-2014 grid at 30-arcsec resolution. From this grid, we selected all pixels between 0 and 200 m below sea level.

From the WDPA, we considered all IUCN recognized PAs (categories I–VI). We obtained 118,609 terrestrial PAs, 2178 marine PAs, and 11,634 coastal (both terrestrial and marine) PAs. For each island, we determined whether there was some protection on terrestrial and marine areas (coded as presence/absence so 1/0) and whether the protection coverage meets the conservation targets of the CDB set for 2020, so 17% for land and 10% for the sea, and the 2030 conservation targets, so 30% coverage on land and sea (coded as achieved/non-achieved so 1/0).

**Climate factors**. We obtained the minimum and maximum temperatures from the WorldClim dataset (version 2), which provides a range of average monthly climate data for 1970–2000 at 30 s (~1 km²) resolution[74]. Minimum and maximum temperatures of each island were calculated by overlaying each of the coolest and warmest months (raster) with the island layer (vector) using the "extract" function from the "raster" package in R-3.4. For each island, we recorded the minimum and maximum rainfall from the driest and wettest month, respectively, in the same manner.

**Geographic and habitat factors**. Island area was calculated directly from the attribute table of island polygons (km²) in ArcGIS 10.3. We also calculated the distance of each island from the continent as the shortest linear distance between this island and the nearest continent in kilometers using the "Near" function in ArcGIS 10.3.

For altitude, each island polygon was intersected with the WorldClim dataset (1 km resolution), using the "Zonal Statistics" function in ArcGIS 10.3 in the same manner as with temperature and rainfall, to extract the highest point, considered as the maximum altitude, and the mean island elevation.

The fractal dimension of each island, as a proxy of tortuosity and habitat diversity, was calculated using the "classStat" function in R-3.4 based on each island shape. The fractal dimension is a normalized measure of polygon shape complexity. Value ranged between 1 (a square) towards 2 (highly convoluted perimeter).

**Land-use and human footprint factors**. To measure freshwater availability as well as the proportion of cropland on each island, we used a raster land cover layer from USGS Land Cover Institute[75]. We first calculated the overall distribution of land cover types per island by overlaying land cover using the "Zonal Histogram" function in ArcGIS 10.3. This function provided a table showing the frequency of categorical cell values (land cover) within each zone (island). From this, we extracted the number and the proportion of cells representing freshwater and cropland, respectively.

We assessed the total human population density per island using the GPWv4 for the year 2015. That grid cell at 30 arcsec (~1 km) consisted in the estimates of human population density based on counts consistent with national censuses and population registers, with cell values representing persons per square kilometer. For each island, we averaged the grid cell values intersecting with its polygon.

**Cultural factors**. We estimated two variables for each island based on the diversity of languages: the total number of languages spoken in the country and the nature of institutional languages. For both variables we used the Ethnologue Global Dataset (20th Edition)[76]. The total number of languages corresponds to the number of living languages in the sovereign state of the island, including both established and immigrant languages. The nature of institutional languages was coded in three categories: institutional languages are only indigenous (I), institutional languages are only non-indigenous (NI), institutional languages mix both indigenous and non-indigenous languages (M).

**Resource capacity factors**. HDI is a synthetic measure capturing elements of life expectancy, education, and wealth. We used HDI 2015 values from the 2016 Human Development Report published by the United Nations of Development program. For each island we manually assigned the HDI value of its sovereign state.

The islands or archipelagos that were part of a continental country, and thus not independent, were considered non-sovereign, whereas the islands and archipelagos that were independent of continental nations were considered as sovereign. This variable was coded as binary (0/1). We had a total of 972 non-sovereign islands out of 2323 (so 41.8%) and 1351 sovereign islands.

**Models**. Prior to model fitting, we evaluated the collinearity between factors using a Pearson's correlation for all islands (Supplementary Fig. 5). Except two pairs of quantitative factors (minimum and maximum temperature, mean and maximum altitude), all factors were weakly or moderately correlated ($-0.7 > rs > +0.7$). We chose to keep all factors and use a backward selection procedure based on AIC to

avoid overfitting. Both minimum and maximum temperature were selected in most of the parsimonious models, while mean and max altitudes were alternatively selected in the most parsimonious model (Table 2). These paired factors also correspond to different hypotheses (Table 1), so deserve to be combined in the full model. Thus, we ran first GLMs with binomial error distribution and a logit link function to predict the probability of PA presence for terrestrial and marine ecosystems on each island using all factors and the "glm" R function. We also ran a similar model to predict the probability of PA coverage >30% on each island for both terrestrial and marine ecosystems.

To take into account potential non-linear relationships between the continuous factors and the coverage by PA on islands we (i) log-transformed all the factors with a large magnitude and asymmetry in their distribution so surface area, human population density, distance to the mainland, and the number of languages, and (ii) included a quadratic term in quantitative factors using the "poly" function in R.

For each model, we computed all subset models (all combinations of factors) using the "dredge" function in R, to calculate the Akaike weight ($w$) that can be interpreted as the probability that a specific model is the best[77]. We then estimated the relative importance of each factor in explaining PA presence or coverage by summing Akaike weight values across all models that include the factor[77]. These summed Akaike weights range from 0 to 1, hence providing a means for ranking the factors in terms of information content. These analyses were performed using the R package "MuMIn."

In addition, we determined the relative dominance of our 16 factors for each binomial GLM based on the $R_E^2$ index[78], which may be interpreted intuitively in a similar way to $R^2$ in the linear regression context. A dominance index compares pairs of factors across all subset models to estimate whether one dominates the other in its conditional contribution. Among many dominance indices, Estrella's $R_E^2$ index satisfies minimum statistical requirements (e.g., unbiased etc.) and is recommended[79].

The most parsimonious models were then selected based on the lowest Akaike AIC (backward procedure using the "stepAIC" function in R) to estimate the significance of each factor ($F$ value), to build partial regression plots while avoiding overfitting. The performances of the most parsimonious models were assessed using the AUC statistic, for which values are considered random when they do not differ from 0.5, poor when they are in the range 0.5–0.7, and useful in the range 0.7–0.9, and excellent >0.9[80].

To test the robustness of our conclusions we also used binomial generalized linear mixed-effects models to add "country" as a random effect in order to allow different intercept values for different countries, thus accounting for country-scale covariates not captured by the contextual data[81].

**Reporting summary**. Further information on research design is available in the Nature Research Reporting Summary linked to this article.

## Data availability

All raw data are available in the Supplementary Dataset 1. GADM database of Global Administrative Areas, version 2.8. Available at: www.gadm.org. Center for International Earth Science Information Network (CIESIN), Columbia University, 2016. Gridded Population of the World, Version 4 (GPWv4): Administrative Unit Center Points with Population Estimates. Palisades, NY: NASA Socioeconomic Data and Applications Center (SEDAC). Available at: https://doi.org/10.7927/H4F47M2C. Accessed 09/10/2017. IUCN and UNEP-WCMC (2016), The World Database on Protected Areas (WDPA) [online], [09/2017]. Cambridge, UK: UNEP-WCMC. Available at: www.protectedplanet.net. The GEBCO_2014 Grid, version 20150318. Available at: http://www.gebco.net. Worldclim 2: New 1-km spatial resolution climate surfaces for global land areas. *International Journal of Climatology*: https://www.worldclim.org/data/bioclim.html. USGS Land Cover Institute: https://archive.usgs.gov/archive/sites/landcover.usgs.gov/globallandcover.html. Ethnologue Global Dataset (20th Edition). 2016 Human Development Report published by the United Nations of Development program (UNDP): http://www.hdr.undp.org.

## Code availability

No custom script was developed, only classical functions from ArcGIS 10.3 and R-3.4.

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

## Acknowledgements

This study was partly funded by the 2015–2016 Belmont Forum and the ANR-BiodivERsA RESERVEBENEFIT project.

## Author contributions

D.M. and M.T. envisioned and led the project. D.M., L.V., A.M., and E.M. carried out most of data collection and analyses, L.V. and E.M. draw the figures. D.M., L.V., E.M., A.M., C.H., J.M., and M.T. contributed to the conceptual ideas, design, analyses, and writing.

## Competing interests

The authors declare no competing interests.
