## [Peer Review File · Nature Communications]

Peer Review File, comments first round:

Reviewer #1 (Remarks to the Author):

General comments

Using a comprehensive global dataset of islands, this study provides insights into the drivers of protected areas and their coverage. The authors conclude that variation in these parameters can be explained by a few key predictor variables, and in particular highlight the importance of human development index and population density.

The strengths of this paper are that it generally uses robust and comprehensive data and that it explores an interesting and relevant suite of predictor variables to better understand why some islands are better protected than others. In this context it is likely to be of broad interest to a range of readers and is an interesting contribution to the field.

However, there are a number of weaknesses that I suggest need to be addressed before it is suitable for publication in Nature Communications. These include:

- Quality of writing. The grammar and use of the correct tense need be improved throughout. I have made several suggestions below, but the paper need a thorough editorial review in this regard to be of an appropriate standard for NC
- The models and statistical analysis need to be better described and reported, particularly with regard to the model formulation and model selection process. I have noted the areas needing the most attention in detailed comments below.
- It's unclear what GAMs weren't used to explore non-linear relationships between the response and predictor variables
- There is no detailed Methods for the section that simulates future changes in HDI and population density. I am also not convinced by this section, particularly as the changes in the changes in these two variables, in isolation from the other predictors, seem a little simplistic.

Detailed Comments

L 71-73 Suggest replace the word "Though" with "However"

P75-77 Suggest replace "few alternatives" with "few exceptions"

L82-84 Poorly written, suggest re-write to : Therefore, island socio-ecosystems can be considered the most vulnerable to biodiversity loss while accounting for up to two-thirds of recent species extinctions."

L84 Suggest delete the word "surprising"

L146 These models are not "predicting" anything in this section. A model selection process has been undertaken to find the model (and suite of predictors) that explains the most variation in the response variable. Suggest this heading be reworded accordingly.

L156-157 I would have thought Generalised Additive Models would have been a more flexible (and efficient) mechanism for exploring non-linear relationships between the response and predictor variables – this should potentially be explored.

Also it is not clear here, or in the Methods, if only the quadratic term was included in the saturated model. Given the repeated reference to 16 predictors, it would be good to see more detail, both here and in the Methods, regarding the formulation of the saturated models and how the inclusion of the quadratic terms was dealt with.

L 163 – I'm not sure that UAC of 0.79 and 0.81 can be characterised as 'accurately predicted'. Suggest rewording this along the lines of "...models performed well with the 10 retained factors - as indicated by relatively high AUC values...".

L195 See comments above re L146

L199 replace "retain" with "retained" and "is accurate" with "performed well". Suggest make the latter change throughout (see also comments above for L163

L202 Replace "are" with "were" and delete the word "only"

L225 I'm not entirely convinced by this section. It's not clear here, or in the Methods, how these

predictions were made. If they were made using the best models (with all predictors) but only changing the values of HDI and population density then that should be clearly explained.

L231 – there are no details of this section in the Methods??

L252 – This is a very unwieldy sentence to start the Discussion. Suggest replace with something like: “We provide a novel global assessment of protected area coverage on islands, utilizing an exhaustive database containing 2,323 island”

L260 – suggest replace “towards”: with “in”

L261-262 – In line with previous comments, suggest it would be more appropriate to phrase this sentence: “...we show that the variation in presence and spatial extent of coverage can be explained with a limited set of factors...”

L460-461 – For remote islands of sovereign states the HDI is likely to be much reduced. While I realise this is hard to quantify this does represent a weakness, and should be addressed in the Discussion

L 471 - A better justification is needed as to why the collinear factors weren't dropped from the model selection process

L472 – As indicated above, the models were used to find the suite of predictors that explained the most variation in the response variables. While the partial regression plots show this relationship for each variable, I still don't think it's appropriate to state that these models were used to predict the probability of the response variables. See also comments above

L483 - Was this using the dredge function? Please be more specific here as it is important for the reader to understand the model selection process used.

L787 – see comments above regarding the details of the selection process – earlier it was stated that all combinations were tested (see L477), which is not a backward selection process

Reviewer #2 (Remarks to the Author):

OVERALL COMMENTS

The study assesses the terrestrial and marine protected area (PA) coverage of islands globally and evaluates a diversity of potential predictor variables to determine to what extent they explain the variation in PA coverage of islands. This is an important topic given the continuing biodiversity loss globally and the vulnerability of island ecosystems in particular. The study is of global scope in an area where relatively little research has provided insights so far (i.e. what explains the variation of PA coverage on islands), making the study a valuable contribution to the literature. However, lack of clarity on a number of points, as outlined below, make it difficult to assess the robustness of the study and would need to be address.

The justification for the study needs to be made more robustly for the study to be more convincing. In framing the study, the authors point towards the positive impacts that PAs have had and some of the international targets that countries have signed up to or that are being proposed. This framing needs to be qualified and looked at through a more critical lens. A considerable literature has shown that PAs can also have negative impacts, especially on marginalised communities, e.g. due to evictions and loss of access to natural resources. There is currently no mention of this in the manuscript and hence, PAs are only shown in a positive light. Furthermore, the authors need to justify better why they look at predictors of PA coverage. Despite drastic increases in PA coverage over the last decades, biodiversity loss continues nearly unabated. It is therefore widely acknowledged that increasing PA coverage is not what is needed to stem biodiversity loss (instead it is key that the ultimate consumption and production drivers of biodiversity loss are addressed). This is not mentioned in the manuscript. This therefore leads the reader to believe that increasing PA coverage will result in enhanced biodiversity conservation (providing a strong reason for better understanding the factors correlated with PA coverage), when this is very unlikely the case. PA coverage targets are set to remain high on the international post-2020 negotiation agenda not because they are particularly effective conservation targets, but because they are relatively easy to achieve and measure. So there are merits in better understanding factors underlying PA coverage, but this needs to be much better caveated in the manuscript.

There is not much explanation of how these specific predictor variables were chosen as opposed to many others that could have been included. This makes it difficult to thoroughly evaluate the robustness of the study. This is a considerable shortcoming as (a) there is a plethora of potential other variables that could be looked at, (b) one would expect to find significant relationships given the large sample size and simply by chance due to the relatively large number of predictor variables being assessed and (c) it gives the impression that the predictor variables are an eclectic mix of variables that were chosen simply because these were the datasets at hand. Without an obvious methodological approach for choosing and eliminating potential variables, there should be a sound theoretical underpinning for including specific variables rather than others. It is important that this is done before conducting the analysis, rather than afterwards. It also be important for the authors to highlight what relationships they would expect to find for each of the selected predictor variables prior to conducting the analyses (i.e. being explicit about the hypotheses).

Linked to these preceding points, it is not clear to me why the authors decided to evaluate the 30% protection target (rather than Aichi target 11). This needs to be justified more clearly. Countries have not signed up to the 30% target, so why would one look at factors that could explain whether countries/islands have reached this target, and specifically why would you expect the predictor variables to play a key role in this decision? It seems to me more likely that other factors (than those assessed) play a more important role in whether islands have reached this target. For example, whether countries have set themselves a 30% protection target or higher (in case countries have done this). Or in cases where islands belong to non-island states that are more heavily populated on the continent, the states might set aside more area of islands to reach their Aichi target 11 because there is more land available for protection on the islands.

The authors also need to be very careful that correlation does not imply causation. While the authors found correlations between certain variables and PA coverage, this does not mean that those variables are the reason for any observed change in PA coverage. The projections that the authors make for future protection based on past relationships seem to imply this. In particular, lines 235-237 and 329-331 (e.g. "we show that a limited increase in HDI would have a significant effect on the presence of protected area") seem to suggest that the authors imply causation based on the observed correlation, which is not justified. The use of the word 'determinants' in the title also suggests this.

SPECIFIC COMMENTS

Abstract

- In the manuscript, the authors generalise in a few places without qualifying their statements. Such as at the beginning of the Abstract: "Islands are biodiversity hotspots" (which is true for many, but not all islands) and "Islands are thus considered as the socio-ecosystems the most vulnerable to species and habitat loss". While it is true that islands are considered amongst the most vulnerable ecosystems, I am not convinced from what is said in the manuscript they are the most vulnerable, especially not to habitat loss.
- The authors say that the variables they have assessed are key in explaining the heterogeneity. It would be more useful to be more specific and say how much variation they explained.
- The Abstract refers to the "30% coverage conservation target" without explaining what this is. This is important given the diverse readership of the journal. Readers not familiar with the details of the conservation debate will not have heard of the proposed 30% coverage target.

Introduction:

- The authors give the abbreviation of protected areas (PAs) at first use, but then use the abbreviation inconsistently.
- It would be beneficial to say explicitly in the introduction why islands are particularly vulnerable given this is the focus of the study (e.g. number of endemics, the geographical isolation of many island has meant that species have not co-evolved with predator introduced by humans, etc.)
- In the last paragraph of the introduction (lines 89-90), you mention the SDGs. This seems out of place here as it's not clear how this is linked to the following outline of the study. A better place would be earlier on (when framing the context of the paper) or in the Discussion.

Results:

- The authors start the results with a long section of ~2 pages explaining the methods. These details should be in the methods not the results section. I acknowledge that it can be difficult given the journal's article layout (where the methods appear at the end) to explain enough methodological context for the reader to understand the results, but it should still not be the purpose of the results to explain the methods in such detail as done here. These details should instead appear in the methods section. The results should only refer to some key points, such as the variables you analysed and what type of analysis you conducted.
- Lines 106-107 state that islands of 10km² minimum were included because of the resolution of the dataset. But some of the key datasets are at 1km², so it's not clear to me how this was used to decide on excluding islands smaller than 10km².
- For some of the predictor variables, it would make sense to control for the size of the islands. This is not discussed, so it's not clear why this was not done.

Methods:

- Governance capacity: I do not think that the variables included here (i.e. HDI and whether the islands are island states) can be referred to as being indicators of 'governance capacity'. To me this is misleading (as there other indicators that would lend themselves better to capturing decision making and institutional strength) and I would suggest to change this heading.
- Human footprint: the same applies here, as population density and cropland area on their own are an incomplete measure of human footprint. To assess human footprint it would be more appropriate to use the human footprint index (Venter et al. 2016 Nat. Commun. 7, 12558), which integrates a diversity of key human impacts.
- The authors seem to equate the 'development' status of a state or island with the Human Development Index, when referring to 'developed islands' (line 219-220). The HDI is a measure of human development at an individual level, rather than at a state/island level.
- Climate factors: I am not clear why the authors chose to include minimum and maximum temperature and precipitation rather than annual averages.
- For these and other predictor variables included reasons for including them should be given more explicitly, including referencing relevant studies or the literature where appropriate.
- A number of the predictor variables operate at the national level. For example, the languages spoken hold true to all islands from a country they belong to. Equally there are many socio-economic and governance factors affecting decisions about PA designations that will similar across PAs within the same country. It could therefore be better to conduct the analysis at the level of the state, rather than having the island as the unit of analysis. Alternatively, states should be included as a factor in the analysis. Reasons for not doing this are not given. There is also no explicit mention of how many states the islands included in the study belong to. This information should be included.
- The authors mention the collinearity between predictor variables. It is not clear to me why the authors to keep in all variables even those that were highly correlated.
- Some key methodological details are omitted that make it difficult to follow what exactly was done and would hinder others to reproduce the work. For example, were the datasets used for the analysis all rasterised and standardized to the same projection, resolution and extent?
- Similarly, I am not sure based on what criteria marine protected areas were attributed to islands. This might have been done based on states and distance from the island. To follow the analysis, it would be important to include such details and any specific cut-offs/criteria used.
- I am surprised to see that the authors include the same variables for marine and terrestrial PAs without mentioning any potential differences. Would one expect the same predictors to apply to marine and terrestrial PAs (an implicit assumptions made by the authors, which is not discussed)?

Reviewer #3 (Remarks to the Author):

Many studies have been done on protected area coverage globally, both terrestrial and marine, although this study brings the novelty of focusing specifically on islands. This brings value since islands tend to have high endemism and have tended to be centers of extinctions in the past, and probably the future. The authors have clearly done some substantial analyses, although I have concerns about the methods and chosen analyses.

The WDPA often has some spatial errors, of varying degrees, and these tend to be more obvious with islands. Did the authors check for the level of these errors or do anything to correct them? What datum was used with the spatial data and how exactly did the authors reproject the various datasets? Properly accounting for both the projection and datum would be essential when dealing with fine scale data such as for islands.

My biggest concern is that I think it is dangerous to directly link the level of development (HDI) with PA coverage in a cause and effect manner as the authors have done. How can you distinguish between more developed yielding more protection and more protection yielding more development? What about all the covarying conditions. I am not convinced by the simulation as currently implemented. It is too simplistic. The world does not progress in such a manner where a single variable can change while all the others stay the same. I recommend dropping this section and focusing on characterizing the existing patterns. The current simulation risks promoting development for humans as a direct way of benefitting conservation, and this is counter to vast numbers of experiences.

Related to the above, one could argue against the conclusion that high population density equal less protection on islands. Hong Kong is an island (although not included here as one) and is among the most densely populated places on the planet, yet it has a very high proportion of its area protected.

Perhaps part of the challenge here is that islands are the data points, in their entirety. Is it realistic to treat all of Madagascar as a single data point? Its protected areas tend to be in the remote mountains, much like happens on continents.

For maximum altitude, a better data source might be SRTM, distributed by CGIAR. They offer versions in 90m and 250m horizontal resolution.

Lines 106 to 107 – what explanatory variables were constraining this minimum area threshold?

Line 183 – change less to fewer

Reviewers' comments:

Reviewer #1 (Remarks to the Author):

General comments

Using a comprehensive global dataset of islands, this study provides insights into the drivers of protected areas and their coverage. The authors conclude that variation in these parameters can be explained by a few key predictor variables, and in particular highlight the importance of human development index and population density.

The strengths of this paper are that it generally uses robust and comprehensive data and that it explores an interesting and relevant suite of predictor variables to better understand why some islands are better protected than others. In this context it is likely to be of broad interest to a range of readers and is an interesting contribution to the field.

Thank you for these positive comments

However, there are a number of weaknesses that I suggest need to be addressed before it is suitable for publication in Nature Communications. These include:

- Quality of writing. The grammar and use of the correct tense need to be improved throughout. I have made several suggestions below, but the paper needs a thorough editorial review in this regard to be of an appropriate standard for NC

We corrected the MS according to your suggestions and we carefully revised the grammar.

- The models and statistical analysis need to be better described and reported, particularly with regard to the model formulation and model selection process. I have noted the areas needing the most attention in detailed comments below.

We now provide more details where needed.

- It's unclear what GAMs weren't used to explore non-linear relationships between the response and predictor variables

Thank you for this suggestion but we preferred GLMs over GAMs for the following reasons:

- GAMs, given their flexibility, have an advantage over GLMs when the goal is to make predictions. By contrast, when the main goal is to explain a given variable and compare factors, GLMs is preferable, because there is a theory behind the coefficients (e.g. proportional increase) making them interpretable. Our study is more about interpretation than prediction, particularly the new version where we removed the section about the simulation of scenarios.
- We took into account non-linearity in relationships between variables by (i) including a quadratic term in quantitative factors and (ii) log-transforming some factors.
- Our analysis weighting the relative dominance of our factors cannot handle GAMs. This analysis is central to our study.

- There is no detailed Methods for the section that simulates future changes in HDI and population density. I am also not convinced by this section, particularly as the changes in these two variables, in isolation from the other predictors, seem a little simplistic.

We removed the simulation section as requested by reviewers.

Detailed Comments

L 71-73 Suggest replacing the word "Though" with "However"

Done

P75-77 Suggest replace “few alternatives” with “few exceptions”

Done

L82-84 Poorly written, suggest re-write to : Therefore, island socio-ecosystems can be considered the most vulnerable to biodiversity loss while accounting for up to two-thirds of recent species extinctions.”

Done

L84 Suggest delete the word “surprising”

Done

L146 These models are not “predicting” anything in this section. A model selection process has been undertaken to find the model (and suite of predictors) that explains the most variation in the response variable. Suggest this heading be reworded accordingly.

Agreed, it now reads “Modelling the presence of protected areas”

L156-157 I would have thought Generalised Additive Models would have been a more flexible (and efficient) mechanism for exploring non-linear relationships between the response and predictor variables – this should potentially be explored.

Discussed before.

Also it is not clear here, or in the Methods, if only the quadratic term was included in the saturated model. Given the repeated reference to 16 predictors, it would be good to see more detail, both here and in the Methods, regarding the formulation of the saturated models and how the inclusion of the quadratic terms was dealt with.

We added more details in the methods. We included the quadratic part that has not been captured by the linear term using the *poly* function in R.

L 163 – I’m not sure that UAC of 0.79 and 0.81 can be characterised as ‘accurately predicted’. Suggest rewording this along the lines of “...models performed well with the 10 retained factors - as indicated by relatively high AUC values....”.

Changed accordingly

L195 See comments above re L146

Done

L199 replace “retain” with “retained” and “is accurate” with “performed well”. Suggest make the latter change throughout (see also comments above for L163)

Done

L202 Replace “are” with “were” and delete the word “only”

Done

L225 I’m not entirely convinced by this section. It’s not clear here, or in the Methods, how these predictions were made. If they were made using the best models (with all predictors) but only changing the values of HDI and population density then that should be clearly explained.

Since all reviewers were not convinced by this section, we removed it.

L231 – there are no details of this section in the Methods??

We removed this section.

L252 – This is a very unwieldy sentence to start the Discussion. Suggest replace with something like: “We provide a novel global assessment of protected area coverage on islands, utilizing an exhaustive database containing 2,323 island”

Done

L260 – suggest replace “towards”: with “in”

Done

L261-262 – In line with previous comments, suggest it would be more appropriate to phrase this sentence: “...we show that the variation in presence and spatial extent of coverage can be explained with a limited set of factors...”

Done

L460-461 – For remote islands of sovereign states the HDI is likely to be much reduced. While I realise this is hard to quantify this does represent a weakness, and should be addressed in the Discussion

We agree, we now discuss this weakness, it reads:

“In our case, the strong positive relationship between HDI and protection coverage is certainly the consequence of the allocation of enough resources to protect insular terrestrial and marine areas knowing that remote islands of sovereign states may certainly have lower HDI than reported at the country scale but this information was not available at the island scale. As an alternative, the decrease of HDI with island remoteness was certainly reflected by the distance to the nearest mainland in our models so accounted for in our partial plots of individual effects conditioned by the others.”

L 471 - A better justification is needed as to why the collinear factors weren't dropped from the model selection process

Done now it reads in the Methods

“Prior to model fitting we evaluated the collinearity between factors using a Pearson correlation for all islands (Fig. S3). Except 2 pairs of quantitative factors (minimum and maximum temperature, mean and maximum altitude) all factors were weakly or moderately correlated ($-0.7 > r_s > +0.7$). We chose to keep all factors and use a backward selection procedure based on AIC to avoid overfitting. Both minimum and maximum temperature were selected in most of the parsimonious models while mean and max altitude were alternatively selected in the most parsimonious model (Table 2). These paired factors also correspond to different hypotheses (Table 1) so deserve to be combined in the full model.”

L472 – As indicated above, the models were used to find the suite of predictors that explained the most variation in the response variables. While the partial regression plots show this relationship for each variable, I still don't think it's appropriate to state that these models were used to predict the probability of the response variables. See also comments above

Corrected accordingly

L483 - Was this using the dredge function? Please be more specific here as it is important for the reader to understand the model selection process used.

Yes, we used the dredge function, we clarified.

L787 – see comments above regarding the details of the selection process – earlier it was stated that all combinations were tested (see L477), which is not a backward selection process

In fact, we did both.

First, we estimated the Akaike weight for all factors using dredge.

Second, we performed a backward selection procedure to draw the partial plots based on a parsimonious model.

We clarified this section in the methods.

Reviewer #2 (Remarks to the Author):

OVERALL COMMENTS

The study assesses the terrestrial and marine protected area (PA) coverage of islands globally and evaluates a diversity of potential predictor variables to determine to what extent they explain the variation in PA coverage of islands. This is an important topic given the continuing biodiversity loss globally and the vulnerability of island ecosystems in particular. The study is of global scope in an area where relatively little research has provided insights so far (i.e. what explains the variation of PA coverage on islands), making the study a valuable contribution to the literature.

Thank you for stressing the novelty of our study.

However, lack of clarity on a number of points, as outlined below, make it difficult to assess the robustness of the study and would need to be address.

The justification for the study needs to be made more robustly for the study to be more convincing. In framing the study, the authors point towards the positive impacts that PAs have had and some of the international targets that countries have signed up to or that are being proposed. This framing needs to be qualified and looked at through a more critical lens. A considerable literature has shown that PAs can also have negative impacts, especially on marginalised communities, e.g. due to evictions and loss of access to natural resources. There is currently no mention of this in the manuscript and hence, PAs are only shown in a positive light.

Exact, we now provide a more balanced view of PAs in the introduction by citing key papers showing the downside of protection efforts. We also added some arguments in the discussion.

Furthermore, the authors need to justify better why they look at predictors of PA coverage. Despite drastic increases in PA coverage over the last decades, biodiversity loss continues nearly unabated. It is therefore widely acknowledged that increasing PA coverage is not what is needed to stem biodiversity loss (instead it is key that the ultimate consumption and production drivers of biodiversity loss are addressed). This is not mentioned in the manuscript. This therefore leads the reader to believe that increasing PA coverage will result in enhanced biodiversity conservation (providing a strong reason for better understanding the factors correlated with PA coverage), when this is very unlikely the case. PA coverage targets are set to remain high on the international post-2020 negotiation agenda not because they are particularly effective conservation targets, but because they are relatively easy to achieve and measure. So there are merits in better understanding factors underlying PA coverage, but this needs to be much better caveated in the manuscript.

We agree that we need to better justify the goal and rational of our study knowing that halting biodiversity loss will need more than PA coverage. We acknowledge that both in the introduction and discussion using the most up-to-date references.

There is not much explanation of how these specific predictor variables were chosen as opposed to

many others that could have been included. This makes it difficult to thoroughly evaluate the robustness of the study. This is a considerable shortcoming as (a) there is a plethora of potential other variables that could be looked at, (b) one would expect to find significant relationships given the large sample size and simply by chance due to the relatively large number of predictor variables being assessed and (c) it gives the impression that the predictor variables are an eclectic mix of variables that were chosen simply because these were the datasets at hand. Without an obvious methodological approach for choosing and eliminating potential variables, there should be a sound theoretical underpinning for including specific variables rather than others. It is important that this is done before conducting the analysis, rather than afterwards. It also be important for the authors to highlight what relationships they would expect to find for each of the selected predictor variables prior to conducting the analyses (i.e. being explicit about the hypotheses).

We now provide a new Table 1 showing the rationale and hypothesis behind the choice of each predictor and the relationship that could be expected with protection coverage, this is a very important point.

Linked to these preceding points, it is not clear to me why the authors decided to evaluate the 30% protection target (rather than Aichi target 11). This needs to be justified more clearly.

We choose the 30% protection target instead of the Aichi target 11 because the latter is a country-scale target, with little meaning for islands which are not states, while the former is more a global coverage target. Moreover, the Aichi target 11 has been shown to not be ambitious enough to protect biodiversity and is under renegotiation while the 30% target is the new milestone towards more ambitious targets to halt biodiversity loss. We better justify this choice in the introduction.

Countries have not signed up to the 30% target, so why would one look at factors that could explain whether countries/islands have reached this target, and specifically why would you expect the predictor variables to play a key role in this decision?

We wanted to avoid a country-based target because many islands are not countries and have no obligation to meet the Aichi target 11. Instead we choose a new and more ambitious target that has recently entered the political agenda via IUCN or other organizations: Synthesis of Views of Parties and Observers on the Scope and Content of the Post-2020 Global Biodiversity Framework (CBD, 2019) or more recently the United Nations (2020) call for protected areas to cover nearly one-third or more of the planet by 2030 as part of an effort to stop a sixth mass extinction and slow global warming.

It seems to me more likely that other factors (than those assessed) play a more important role in whether islands have reached this target.

Possibly but our set of factors provided a convincing explanation with high AUC values. We do not claim anymore these factors are causal but they are clearly strong correlates of protection coverage.

For example, whether countries have set themselves a 30% protection target or higher (in case countries have done this). Or in cases where islands belong to non-island states that are more heavily populated on the continent, the states might set aside more area of islands to reach their Aichi target 11 because there is more land available for protection on the islands.

Exact, this is one of the reasons why sovereignty and population density were included as factors, we did the same work for all factors in the new Table 1.

The authors also need to be very careful that correlation does not imply causation. While the authors found correlations between certain variables and PA coverage, this does not mean that those variables are the reason for any observed change in PA coverage. The projections that the authors make for future protection based on past relationships seem to imply this. In particular, lines 235-237 and 329-331 (e.g. "we show that a limited increase in HDI would have a significant effect on the presence of protected area") seem to suggest that the authors imply causation based on the observed correlation, which is not justified. The use of the word 'determinants' in the title also suggests this.

We agree, we changed 'determinants' to 'correlates' in the title, we also removed the last section about scenarios.

SPECIFIC COMMENTS

Abstract

- In the manuscript, the authors generalise in a few places without qualifying their statements. Such as at the beginning of the Abstract: "Islands are biodiversity hotspots" (which is true for many, but not all islands) and "Islands are thus considered as the socio-ecosystems the most vulnerable to species and habitat loss". While it is true that islands are considered amongst the most vulnerable ecosystems, I am not convinced from what is said in the manuscript they are the most vulnerable, especially not to habitat loss.

We added more arguments and references in the introduction on that point. We also modified the abstract substantially.

- The authors say that the variables they have assessed are key in explaining the heterogeneity. It would be more useful to be more specific and say how much variation they explained.

The importance of each factor is reported in Figure 2 and the new Table 2.

- The Abstract refers to the "30% coverage conservation target" without explaining what this is. This is important given the diverse readership of the journal. Readers not familiar with the details of the conservation debate will not have heard of the proposed 30% coverage target.

We changed the abstract to address this issue.

Introduction:

- The authors give the abbreviation of protected areas (PAs) at first use, but then use the abbreviation inconsistently.

Fixed

- It would be beneficial to say explicitly in the introduction why islands are particularly vulnerable given this is the focus of the study (e.g. number of endemics, the geographical isolation of many island has meant that species have not co-evolved with predator introduced by humans, etc.)

We now provide more quantitative arguments and more references, particularly about predators.

- In the last paragraph of the introduction (lines 89-90), you mention the SDGs. This seems out of place here as it's not clear how this is linked to the following outline of the study. A better place would be earlier on (when framing the context of the paper) or in the Discussion.

Done

Results:

- The authors start the results with a long section of ~2 pages explaining the methods. These details should be in the methods not the results section. I acknowledge that it can be difficult given the journal's article layout (where the methods appear at the end) to explain enough methodological context for the reader to understand the results, but it should still not be the purpose of the results to explain the methods in such detail as done here. These details should instead appear in the methods section. The results should only refer to some key points, such as the variables you analysed and what type of analysis you conducted.

We decreased the length of the methods in the results.

- Lines 106-107 state that islands of 10km² minimum were included because of the resolution of the

dataset. But some of the key datasets are at 1km², so it's not clear to me how this was used to decide on excluding islands smaller than 10km².

In fact, we excluded uninhabited islands and those which had no value for factors like those at high absolute latitudes or being too small.

- For some of the predictor variables, it would make sense to control for the size of the islands. This is not discussed, so it's not clear why this was not done.

Island area/size is on the factors considered in this study, so when we provide partial plots for all other factors we control for island size/area. We also assess island size effect by itself (Table 2) and we show its influence on the presence of PA when significant (Figure S1 & S2). We also discuss this factor in the new Table 1.

Methods:

- Governance capacity: I do not think that the variables included here (i.e. HDI and whether the islands are island states) can be referred to as being indicators of 'governance capacity'. To me this is misleading (as there other indicators that would lend themselves better to capturing decision making and institutional strength) and I would suggest to change this heading.

We agree, we propose instead to refer to resource capacity. Governance capacity indeed incorporates more than what we include here (e.g. indicators of decision making and institutional strength). However, we can say that a country HDI and sovereignty status can still provide an indicator of the extent to which a country is likely to have access to resources (e.g. human & financial capital) that increase their capacity to manage their natural environment. We removed reference to governance in the text and figures.

- Human footprint: the same applies here, as population density and cropland area on their own are an incomplete measure of human footprint. To assess human footprint it would be more appropriate to use the human footprint index (Venter et al. 2016 Nat. Commun. 7, 12558), which integrates a diversity of key human impacts.

We looked at this option but human footprint is a composite index which highly correlated to human density in our case (>0.9). We prefer to keep individual variables like cropland area and human density instead of using a composite one which is harder to interpret.

- The authors seem to equate the 'development' status of a state or island with the Human Development Index, when referring to 'developed islands' (line 219-220). The HDI is a measure of human development at an individual level, rather than at a state/island level.

Exact, we do not refer anymore to 'developed' vs. 'developing' countries or islands but islands belonging to countries with high or low HDI.

- Climate factors: I am not clear why the authors chose to include minimum and maximum temperature and precipitation rather than annual averages.

We now provide a rationale and hypotheses for each factor in Table 1. We know that minimum and maximum variables are often more informative than the mean. They are selected in most parsimonious models showing their importance and complementarity (Table 2, Figure 2).

- For these and other predictor variables included reasons for including them should be given more explicitly, including referencing relevant studies or the literature where appropriate.

We now provide a rationale and hypotheses for each factor in Table 1.

- A number of the predictor variables operate at the national level. For example, the languages spoken hold true to all islands from a country they belong to. Equally there are many socio-economic and governance factors affecting decisions about PA designations that will be similar across PAs within the

same country. It could therefore be better to conduct the analysis at the level of the state, rather than having the island as the unit of analysis. Alternatively, states should be included as a factor in the analysis. Reasons for not doing this are not given.

In fact, since many factors are state-based values we cannot include the state as factor, it would be redundant. Since many factors can be assessed at the island scale we prefer to keep this spatial grain.

There is also no explicit mention of how many states the islands included in the study belong to. This information should be included.

Sure

We had a total of 101 countries with a mean of 23 islands per country (SD=60 islands), and 1351 sovereign islands out of 2,323.

We included that information in the main text.

- The authors mention the collinearity between predictor variables. It is not clear to me why the authors to keep in all variables even those that were highly correlated.

We just removed factors that were highly correlated like human footprint and human density (>0.9) then we made the choice to perform a selection procedure without any *a priori* on the final list of variables to select.

- Some key methodological details are omitted that make it difficult to follow what exactly was done and would hinder others to reproduce the work. For example, were the datasets used for the analysis all rasterised and standardized to the same projection, resolution and extent?

We used global spatial layers (shapefile and raster layers) that are freely available online. We now provide detailed references in the "Methods" section. Global raster layers at 30 seconds (~1 km²) were used. All the datasets were converted to the Behrmann (world) projected coordinate system for quantitative analyses.

We now added more information in the methods.

- Similarly, I am not sure based on what criteria marine protected areas were attributed to islands. This might have been done based on states and distance from the island. To follow the analysis, it would be important to include such details and any specific cut-offs/criteria used.

We now added more information in the methods.

In fact, we estimated the proportion of marine area under protection for a given island, as the part of the continental shelf in the territorial waters (12 nautical miles) which intersected with the World Database on Protected Areas (WDPA) on January 2019. Continental shelf has been extracted from the GEBCO-2014 grid at 30-arc second resolution. From this grid we selected all pixels between 0 and 200 meters below sea level.

- I am surprised to see that the authors include the same variables for marine and terrestrial PAs without mentioning any potential differences. Would one expect the same predictors to apply to marine and terrestrial PAs (an implicit assumptions made by the authors, which is not discussed)?

Yes, most of the time the same criteria apply but for different reasons that are now explicitly described in our new Table 1.

Reviewer #3 (Remarks to the Author):

Many studies have been done on protected area coverage globally, both terrestrial and marine, although this study brings the novelty of focusing specifically on islands. This brings value since islands tend to have high endemism and have tended to be centers of extinctions in the past, and probably the future. The authors have clearly done some substantial analyses, although I have concerns about the methods and chosen analyses.

The WDPA often has some spatial errors, of varying degrees, and these tend to be more obvious with islands. Did the authors check for the level of these errors or do anything to correct them? What datum was used with the spatial data and how exactly did the authors reproject the various datasets? Properly accounting for both the projection and datum would be essential when dealing with fine scale data such as for islands.

Indeed, we had to check all MPA polygons one by one to avoid marine protected areas delineating only terrestrial systems or terrestrial protected areas encompassing a large coastal area.

To determine the percentage of protection for each individual island, we estimated the proportion of terrestrial and marine area, the latter defined as the part of the continental shelf in the territorial waters (12 nautical miles), which intersected with the World Database on Protected Areas (WDPA) on January 2019. Continental shelf has been extracted from the GEBCO-2014 grid at 30-arc second resolution. From this grid we selected all pixels between 0 and 200 meters below sea level.

We now added more information in the methods.

My biggest concern is that I think it is dangerous to directly link the level of development (HDI) with PA coverage in a cause and effect manner as the authors have done. How can you distinguish between more developed yielding more protection and more protection yielding more development? What about all the covarying conditions. I am not convinced by the simulation as currently implemented. It is too simplistic. The world does not progress in such a manner where a single variable can change while all the others stay the same. I recommend dropping this section and focusing on characterizing the existing patterns. The current simulation risks promoting development for humans as a direct way of benefitting conservation, and this is counter to vast numbers of experiences.

We agree with the reviewer and we dropped the scenario/simulation section. Elsewhere we paid more attention to not use causality instead of correlation, including in the title. We also propose a more balanced view of PAs in the introduction and discussion including the bad side of them.

Related to the above, one could argue against the conclusion that high population density equal less protection on islands. Hong Kong is an island (although not included here as one) and is among the most densely populated places on the planet, yet it has a very high proportion of its area protected.

This counter-example shows the limitation of the causality approach. It also highlights the complexity and multifactorial aspects of protection coverage; no single factor can explain a substantial part of variation. We added that remark in the discussion.

Perhaps part of the challenge here is that islands are the data points, in their entirety. Is it realistic to treat all of Madagascar as a single data point? Its protected areas tend to be in the remote mountains, much like happens on continents.

Yes, we study entire islands so we averaged values. The remoteness of some parts of Madagascar can be approximated by fractal dimension, maximum altitude or population density. We now provide a new Table 1 showing rationale and hypotheses linking each factor to protection coverage.

For maximum altitude, a better data source might be SRTM, distributed by CGIAR. They offer versions in 90m and 250m horizontal resolution.

We checked this alternative and since there is not substantial improvement at the scale of entire islands we prefer to keep the initial WorldClim dataset at 1km resolution which seems enough at the

island scale.

Lines 106 to 107 – what explanatory variables were constraining this minimum area threshold?

Land use for instance with a surface area for each category is a problem at very fine resolution. This minimum is also driver by the selection of inhabited islands only. This was better justified in the methods.

Line 183 – change less to fewer

Done

Reviewers' comments: second round

Reviewer #1 (Remarks to the Author):

The authors have done a thorough job of addressing all my comments and suggested changes. The revised manuscript is much improved. The findings are novel and of broad interest.

Aleks Terauds

Reviewer #2 (Remarks to the Author):

The revisions have improved the manuscript considerably and have addressed the majority of the points I had raised when reviewing the previous version. Many thanks to the authors for their work.

There are a few outstanding points that I think need to be addressed to make it suitable for potential publication.

Providing a more balanced views on protected areas (PAs):

The authors have added a few sentences and relevant publications highlighting potential negative impacts and limitations of PAs. These revisions are very welcome. However, in my view the manuscript is still not sufficiently balanced to reflect the evidence base on PAs. In particular, a few sentences have not been revised to reflect the changes made on this to the manuscript and continue to suggest that increasing PA coverage is needed to address the environmental crises.

For example, line 103 to 104 mentions such targets to be "a necessary first step in galvanizing action", which is contradictory to the first part of the same sentence and earlier mention of the limitations of PAs. This second part of the sentence should therefore be deleted (or revised). In addition, lines 66-67 and lines 335-337 still reflect the previous assumption of the manuscript that increasing PA coverage is needed and will result in better biodiversity conservation. This is not in line with the references now cited and the limitations of PAs mentioned elsewhere in the manuscript. These sentence should therefore be revised to make more explicit that increasing PA coverage can have positive environmental impacts, but this will not necessarily be the case. Lines 61 to 63 now reference some key limitations of PAs. I suggest to also add here that the human pressure can also simply be displaced from inside PAs to elsewhere (i.e. lead to so called leakage).

Amongst the references cited here (lines 61-66), it would be most valuable to reference a literature review on the impacts of PAs (rather than primarily referencing case studies), such as by Oldekop et al. (2016; A global assessment of the social and conservation outcomes of protected areas. *Conserv. Biol.* 30, 133–141).

The Abstract needs to be revised further:

Some of the sentences need to be qualified. In particular, the first sentence states that "Islands are biodiversity hotspots" – this is not the case for all islands, so a wording such as "Many islands are biodiversity hotspots" would be more appropriate.

The last sentence of the Abstract is left from the sections now deleted from the manuscript making projections into the future and is no longer supported by the main text. The sentence should be deleted.

Results section:

The authors have cut down the methods mentioned in the results section considerably, which is a step in the right direction. However, there are still too many details on methods mentioned in the results section. The authors should only mention the absolute necessary to make sense of the results. This means omitting many of the details in the Results and moving those to the Methods, such as lines 128, 171 to 174, 184-185, and 210-212.

Methods:

I am not convinced by the reasons given by the authors for not accounting for the state being the level at which data are available for a number of the datasets. Just because many variables are available at finer resolution, one cannot use a data analysis approach for the data where the finer resolution data aren't available (i.e. where you only have state level data).

It would be a much more robust approach to include state as random effect (e.g. such as done in a similar context by Geldmann et al. 2018. Conservation Letters 11:e12434; or use an alternative approach consistent with the unit of analysis being the state where this is the case). This means that where data are available at a finer resolution (i.e. for each island) the analysis would account for this and be consistent with what the authors want to achieve while not assuming a finer resolution unit of analysis where that isn't the case.

At the very least, the authors should conduct a sensitive analysis repeating the analysis with state as a random effect, report those results and how sensitive the results are to this, as well as clearly highlight any resulting limitations in the manuscript.

On a separate point, the definition of PAs given at the beginning of the manuscript (lines 57-58) is not the same as the one of the IUCN. Given that you use the WDPA and all IUCN PA categories as the PA data in your analysis, you should be using the same definition of PAs.

A minor point that is confusing: in the results section you mention that Greenland was excluded (line 124) citing the Methods section. However, in the methods you only mention that Greenland was included (line 358).

Improving the readability of the manuscript:

The manuscript would still benefit from a thorough review of the use of grammar and consistency of language throughout. For example, an article is missing in front of SDGs in line 59, 'behavior' should be plural in line 83 (or an article should be inserted), and 'islands' are sometimes written with a capital 'I', but not always (e.g. see line 247).

As previously mentioned, acronyms are not used consistently throughout (e.g. protected areas are spelled out in some place; and in particular lines 66, 172, 175, 318, 329, 330, 335, 348, and 387 need revising).

In addition, the sentence covering lines 158 to 162 is very long and therefore difficult to follow. It is also not clear to me what 'group membership' is referring to in line 285 and reference 25 is incomplete.

Reviewer #3 (Remarks to the Author):

The revised manuscript is certainly clearer than the original, but there are fundamental issues of concern, some that are now more obvious. To not waste time and get straight to the point, the changing wording along the manuscript regarding the response variable is confusing. Is the response simple presence or absence of protection, and never the proportion protected? In that case, Madagascar having one terrestrial and one marine PA, covering some tiny fraction of the total area, would make it yes in all models. The same would be the case for a small island with 99% coverage of its terrestrial and marine areas. Am I understanding that correctly? That seems an inadequate approach to the questions and would make many of the current conclusions misleading. To me, this would be a fatal flaw in the current analyses, although one that is correctable.

I understand why the model for achieving the 30% target has a binary response, but for understanding possible drivers of protection rates, the response should be proportion of the island or marine areas protected. Simple presence/absence is not all that interesting. Given how much of the manuscript is talking about rates of protection, I do not understand why one would even model simple presence of protection. This is perhaps why I did not appreciate this in my first review. It is

not clear if the other reviewers fully appreciated this either, based on their comments.

My other major concern is the input variables and the rationales for them, now in Table 1. Many of the justifications in Table 1 could arguably work against using the listed variables. For example, all the temperature and precipitation variables are justified by their influence on species numbers. If the justification for climate variables is their effect on species richness, then heterogeneity or range of values would make more sense (e.g., temperature interval instead of min and max). Regardless, you could use species numbers directly, which you could estimate, at least for the terrestrial vertebrates or marine species having global databases. As a counter example, distance from the mainland is also likely to affect the number of species, but in this case, it is not even mentioned as a rationale. Overall, I do not find the rationales convincing. Table 1 also has many grammar errors.

Minor issues

Lines 129-130, These statistics are confusing. Is it the size of island fully within a PA, or islands having some part of their area with a PA? I would imagine nearly all islands are only partially within PAs, since your analysis, by definition, looks only at inhabited islands.

Line 132, Islands should be islands. This error occurs in other places too.

Line 144, delete "highly". Lower p-values do not make the result more significant. It just passes a more stringent test, and it is largely a function of having a large sample.

In the abstract, line 40, it might be more appropriate to say, "We show that, on average, 22% of terrestrial and 13% of marine island 'area is within protected areas'." We know that just because a geographic area is within a polygon doesn't always mean there is any real protection.

Reviewer #1 (Remarks to the Author):

The authors have done a thorough job of addressing all my comments and suggested changes. The revised manuscript is much improved. The findings are novel and of broad interest.

Thank you for this positive re-assessment.

Aleks Terauds

Reviewer #2 (Remarks to the Author):

The revisions have improved the manuscript considerably and have addressed the majority of the points I had raised when reviewing the previous version. Many thanks to the authors for their work.

We indeed entirely revised our first version.

There are a few outstanding points that I think need to be addressed to make it suitable for potential publication.

Providing a more balanced views on protected areas (PAs):

The authors have added a few sentences and relevant publications highlighting potential negative impacts and limitations of PAs. These revisions are very welcome. However, in my view the manuscript is still not sufficiently balanced to reflect the evidence base on PAs. In particular, a few sentences have not been revised to reflect the changes made on this to the manuscript and continue to suggest that increasing PA coverage is needed to address the environmental crises.

We are happy to follow Reviewer's suggestion by providing an even more balanced view on PAs and more relevant references (lines 68-73).

For example, line 103 to 104 mentions such targets to be "a necessary first step in galvanizing action", which is contradictory to the first part of the same sentence and earlier mention of the limitations of PAs. This second part of the sentence should therefore be deleted (or revised).

We agree, it now reads: "these targets and measures of achievements will likely remain and be complemented by evaluations of the broader benefits of protected areas (Visconti et al 2019 Science).

Visconti, P., Butchart, S.H., Brooks, T.M., Langhammer, P.F., Marnewick, D., Vergara, S., Yanosky, A. and Watson, J.E., 2019. Protected area targets post-2020. *Science*, 364(6437), pp.239-241.

In addition, lines 66-67 and lines 335-337 still reflect the previous assumption of the manuscript that increasing PA coverage is needed and will result in better biodiversity conservation. This is not in line with the references now cited and the limitations of PAs mentioned elsewhere in the manuscript. These sentence should therefore be revised to make more explicit that increasing PA coverage can have positive environmental impacts, but this will not necessarily be the case.

Done

Lines 61 to 63 now reference some key limitations of PAs. I suggest to also add here that the human pressure can also simply be displaced from inside PAs to elsewhere (i.e. lead to so called leakage).

Important point, thank you. Done

Amongst the references cited here (lines 61-66), it would be most valuable to reference a literature review on the impacts of PAs (rather than primarily referencing case studies), such as by Oldekop et al. (2016; A global assessment of the social and conservation outcomes of protected areas. *Conserv. Biol.* 30, 133–141).

We added Oldekop et al. (2016) which is indeed very relevant.

The Abstract needs to be revised further:

Some of the sentences need to be qualified. In particular, the first sentence states that “Islands are biodiversity hotspots” – this is not the case for all islands, so a wording such as “Many islands are biodiversity hotspots” would be more appropriate.

Done

The last sentence of the Abstract is left from the sections now deleted from the manuscript making projections into the future and is no longer supported by the main text. The sentence should be deleted.

Done, it now reads “Our study suggests that economic development and population growth may critically limit the amount of protection on islands”.

Results section:

The authors have cut down the methods mentioned in the results section considerably, which is a step in the right direction. However, there are still too many details on methods mentioned in the results section. The authors should only mention the absolute necessary to make sense of the results. This means omitting many of the details in the Results and moving those to the Methods, such as lines 128, 171 to 174, 184-185, and 210-212.

We removed more methods in the results as suggested.

Methods:

I am not convinced by the reasons given by the authors for not accounting for the state being the level at which data are available for a number of the datasets. Just because many variables are available at finer resolution, one cannot use a data analysis approach for the data where the finer resolution data aren't available (i.e. where you only have state level data).

It would be a much more robust approach to include state as random effect (e.g. such as done in a similar context by Geldmann et al. 2018. Conservation Letters 11:e12434; or use an alternative approach consistent with the unit of analysis being the state where this is the case). This means that where data are available at a finer resolution (i.e. for each island) the analysis would account for this and be consistent with what the authors want to achieve while not assuming a finer resolution unit of analysis where that isn't the case.

At the very least, the authors should conduct a sensitive analysis repeating the analysis with state as a random effect, report those results and how sensitive the results are to this, as well as clearly highlight any resulting limitations in the manuscript.

Most socio-economic variables are only available at the island scale. Since some islands are also countries we have no replicates. Nevertheless, we ran mixed models (GLMM) with the country as a random effect as suggested to allow different intercept values for different countries. All these results are presented in a new appendix (Figures S2 & S4, Table S1). They are very consistent with those obtained with fixed-effects models. It shows that within each country, islands can have individual variations given environmental and human conditions like for inter-country patterns. Unsurprisingly, the 4 country-scale factors have lower AIC weights with mixed models. To keep the effects of country-scale factors we prefer to keep the GLM model outputs in the main text.

On a separate point, the definition of PAs given at the beginning of the manuscript (lines 57-58) is not the same as the one of the IUCN. Given that you use the WDPA and all IUCN PA categories as the PA data in your analysis, you should be using the same definition of PAs.

Corrected, it now reads "Protected areas (PAs) are clearly defined geographical spaces, recognized, dedicated, and managed through legal or other effective means to achieve the long-term conservation of nature with associated ecosystem services and cultural values (IUCN 2008)"

A minor point that is confusing: in the results section you mention that Greenland was excluded (line 124) citing the Methods section. However, in the methods you only mention that Greenland was included (line 358).

Sorry for the contradiction, Greenland is in fact excluded due to a lack of data at high latitude.

Improving the readability of the manuscript:

The manuscript would still benefit from a thorough review of the use of grammar and consistency of language throughout. For example, an article is missing in front of SDGs in line

59, 'behavior' should be plural in line 83 (or an article should be inserted), and 'islands' are sometimes written with a capital 'I', but not always (e.g. see line 247).

Done, a careful reading was also carried out.

As previously mentioned, acronyms are not used consistently throughout (e.g. protected areas are spelled out in some place; and in particular lines 66, 172, 175, 318, 329, 330, 335, 348, and 387 need revising).

Corrected

In addition, the sentence covering lines 158 to 162 is very long and therefore difficult to follow.

We cut the sentence.

It is also not clear to me what 'group membership' is referring to in line 285 and reference 25 is incomplete.

We have edited this text to more accurately represent the literature. It now reads:
"Where both indigenous and non-indigenous institutional languages co-exist, barriers to communication may exist impacting a country's institutional performance"

Ref #25 has been completed

Reviewer #3 (Remarks to the Author):

The revised manuscript is certainly clearer than the original, but there are fundamental issues of concern, some that are now more obvious. To not waste time and get straight to the point, the changing wording along the manuscript regarding the response variable is confusing. Is the response simple presence or absence of protection, and never the proportion protected?

Sorry for the confusion.

We considered two binary response variables (1/0):

- (i) the presence (1) /absence (0) of protection (independently of the protection coverage);
- (ii) the success (1) vs the failure (0) in achieving the 30% protection target.

For both binary response variables, we used binominal GLMs and the same set of 16 social and environmental variables.

We clarified our objectives where it reads:

"Finally, to understand how social and environmental characteristics are associated with PA coverage on islands, we modelled (i) the probability of terrestrial and marine PA presence on

each island, and (ii) the likelihood that each island already meets the new global target of 30% coverage on both terrestrial and marine areas.”

We did not predict the proportion detected for two main reasons:

- The distribution of the proportion of area protected is U-shaped with many 0 and 1 and few intermediate values. Fitting such distribution is also challenging so we prefer to fit a binomial response.
- Our goal is not to find variables explaining the difference between islands having, let say 60% of protection coverage and 90%, but to model the likelihood to reach a certain threshold (0% or 30%).

Lines 160-162 it now reads: “We used binomial generalized linear models (GLMs) to explore how 16 social and environmental factors are associated with the presence of protection, independently of the coverage, on terrestrial and marine areas on each island (binary response variable yes/no).”

In that case, Madagascar having one terrestrial and one marine PA, covering some tiny fraction of the total area, would make it yes in all models.

Exactly, if at least one terrestrial or marine PA is present the response in the first model (presence/absence of protection) is true (1).

We also built a second model to predict the success (1) /failure (0) in achieving the 30% protection coverage target given social and environmental conditions.

The same would be the case for a small island with 99% coverage of its terrestrial and marine areas. Am I understanding that correctly?

Exact, for the first model focusing only on presence/absence of protection 1% and 99% coverage would provide the same response which is presence = 1. However, the second model focusing on the success/failure of achieving the 30% coverage target would discriminate 1% coverage as failure (0) and 99% coverage as success (1).

Above 30% coverage, we consider that modelling differences among islands was less important.

That seems an inadequate approach to the questions and would make many of the current conclusions misleading. To me, this would be a fatal flaw in the current analyses, although one that is correctable.

We hope our clarification overcomes the problem.

I understand why the model for achieving the 30% target has a binary response, but for understanding possible drivers of protection rates, the response should be proportion of the island or marine areas protected.

In fact, we modelled the success or failure to reach this 30% target. We do not try to explain the proportion covered since we are not seeking why some islands have 60% vs. 80% coverage by protection.

Simple presence/absence is not all that interesting. Given how much of the manuscript is talking about rates of protection, I do not understand why one would even model simple presence of protection. This is perhaps why I did not appreciate this in my first review. It is not clear if the other reviewers fully appreciated this either, based on their comments.

We now clarify and justify this choice in the introduction lines 121-123.

“Above 30% coverage, we considered that modelling differences among islands was of less importance. Our main goal was not to focus on high values of protection coverage (30-100%) but to investigate the correlates of achievement of different thresholds corresponding to policy targets.”

Also in the discussion lines 247-248

“We did not model this U-shape distribution to avoid focusing on extreme high values but on certain thresholds that correspond to policy targets.”

My other major concern is the input variables and the rationales for them, now in Table 1. Many of the justifications in Table 1 could arguably work against using the listed variables. For example, all the temperature and precipitation variables are justified by their influence on species numbers. If the justification for climate variables is their effect on species richness, then heterogeneity or range of values would make more sense (e.g., temperature interval instead of min and max). Regardless, you could use species numbers directly, which you could estimate, at least for the terrestrial vertebrates or marine species having global databases.

We understand the comment but species richness is not available at the scale of most islands which are very small while species ranges are large and do not provide information at the scale of small islands. Moreover, using only vertebrates appears limited.

Since our models provide accurate predictions we consider that our set of environmental, geographic and socioeconomic variables embeds relevant correlates, and not causal factors, of protection coverage. Using more correlates would be useful in case of poor prediction which is not the case.

As a counter example, distance from the mainland is also likely to affect the number of species, but in this case, it is not even mentioned as a rationale. Overall, I do not find the rationales convincing. Table 1 also has many grammar errors.

We revised our Table 1 to provide more arguments and correct some errors.

The well-known theory of biogeography was omitted while this is the main driver of species richness on islands.

Minor issues

Lines 129-130, These statistics are confusing. Is it the size of island fully within a PA, or islands having some part of their area with a PA? I would imagine nearly all islands are only partially within PAs, since your analysis, by definition, looks only at inhabited islands.

We clarified this section.

Line 132, Islands should be islands. This error occurs in other places too.

Corrected

Line 144, delete “highly”. Lower p-values do not make the result more significant. It just passes a more stringent test, and it is largely a function of having a large sample.

Corrected

In the abstract, line 40, it might be more appropriate to say, “We show that, on average, 22% of terrestrial and 13% of marine island ‘area is within protected areas’.” We know that just because a geographic area is within a polygon doesn’t always mean there is any real protection.

We follow your suggestion it now reads “We show that, on average, 22% of terrestrial and 13% of marine island areas are covered by protection status”

OUTSTANDING REVIEWERS' CONCERNS:

The reviewers provided their latest comments only as remarks to the editors, therefore I summarise them here. Reviewer 3 maintained that you and your co-authors could have addressed a more interesting and relevant question following the alternative analytical approach suggested. However, Reviewer 3 was open to not letting this judgement "sink" your paper if the other referees and the editors were of a different opinion. We therefore consulted another reviewer on this matter. After giving the issue considerable thought, this reviewer agreed that your approach is appropriate from the perspective of testing hypotheses that provide insights into the conservation planning processes – even though they found your reasoning in the response to Reviewer 3 somewhat weak (e.g. there are a number of modelling approaches that could deal with the challenging distributions of the type you outlined). The reviewer suggests that undertaking the modelling recommendations by Reviewer 3 and including them in the Supplementary Information would strengthen the paper.

I discussed the matter with the other editors, and we decided to ask that you consider adding those additional analyses as Supplementary Information. If you feel this is out of scope or unfeasible in a short time, at least please discuss the alternative approach in the manuscript, along with a caveat about the limitations of the current approach.